# COOPERATIVE MULTI-AGENT RL WITH COMMUNICATION CONSTRAINTS

## ABSTRACT

Cooperative Multi-agent reinforcement learning (MARL) often assumes frequent access to global information in a data buffer, such as team rewards or other agents' actions, which is typically unrealistic in decentralized MARL systems due to high communication costs. When communication is limited, agents must rely on outdated information to estimate gradients and update their policies. A common approach to handle missing data is called importance sampling, in which we reweigh old data from a base policy to estimate gradients for the current policy. However, it quickly becomes unstable when the communication is limited (i.e. missing data probability is high), so that the base policy in importance sampling is outdated. To address this issue, we propose a technique called *base policy prediction*, which utilizes old gradients to predict the policy update and collect samples for a sequence of base policies, which reduces the gap between the base policy and the current policy. This approach enables effective learning with significantly fewer communication rounds, since the samples of predicted base policies could be collected within one communication round. Theoretically, we show that our algorithm converges to an $\varepsilon$-Nash equilibrium in potential games with only $\mathcal{O}(\varepsilon^{-3/4})$ communication rounds and $\mathcal{O}(\mathrm{poly}(\max_i |\mathcal{A}_i|) \cdot \varepsilon^{-11/4})$ samples, improving existing state-of-the-art results in communication cost, as well as sample complexity without the exponential dependence on the joint action space size. We also extend these results to general Markov Cooperative Games to find an agent-wise local maximum. Empirically, we test the base policy prediction algorithm in both simulated games and MAPPO for complex environments. The results show that our algorithms can significantly reduce the communication costs while maintaining good performance compared to the setting without communication constraints, and standard algorithms fail under the same communication cost.

## 1 INTRODUCTION

In recent years, multi-agent reinforcement learning (MARL) has achieved remarkable empirical success. A large number of MARL applications consider fully cooperative settings, where all agents share a common goal and aim to maximize the joint team reward (Yu et al., 2022; Rashid et al., 2020; Sunehag et al., 2017). In such scenarios, the challenge is to enable effective coordination to jointly optimize the reward together, cope with exponential joint action space and dynamic environments in order to achieve optimal performance. To achieve this goal, many papers assume that all agents can get access to the team reward and other agents' actions at each time, hence each agent can update their policies accordingly.

However, they often ignore the cost of getting team rewards and other agents' actions. In the real world, each agents usually can only observe partial rewards, while getting global information requires communication with each other. Or, all agents can only get global information by accessing the replay buffer that can store all data. In both scenarios, getting global information could lead to a huge communication cost in a large-scale decentralized MARL system, such as multi-robot systems (Orr and Dutta, 2023), sensor networks (Zhao et al., 2023), or traffic control (Chu et al., 2019). For example, in a large-scale robot system to transport packages, many mobile robots are coordinated using MARL (Shen et al., 2023). However, requiring each individual agent to obtain global information at every timestep incurs substantial communication overhead, which will significantly slow

Table 1: The Comparisons between representative works in both Potential Games and Markov Co-operative Games. Terms other than $|\mathcal{A}_i|$ and $\varepsilon$ are ignored.

|  | Sample Complexity | Communication Cost | Equilibrium Type |
|---|---|---|---|
| Song et al. (2021) | $\mathcal{O}(\max_{i \in [n]} |\mathcal{A}_i|/\varepsilon^3)$ | $\mathcal{O}(\max_{i \in [n]} |\mathcal{A}_i|/\varepsilon^3)$ | Nash Equilibrium |
| Liu et al. (2021) | $\mathcal{O}(\prod_{i=1}^n |\mathcal{A}_i|/\varepsilon^2)$ | $\mathcal{O}(\prod_{i=1}^n |\mathcal{A}_i|/\varepsilon^2)$ | Nash Equilibrium |
| (Leonardos et al., 2021) (Ding et al., 2022) | $\mathcal{O}(\max_{i \in [n]} |\mathcal{A}_i|/\varepsilon^5)$ | $\mathcal{O}(\max_{i \in [n]} |\mathcal{A}_i|/\varepsilon^5)$ | Nash Equilibrium |
| Wang et al. (2023) | $\mathcal{O}(\max_{i \in [n]} |\mathcal{A}_i|/\varepsilon^2)$ | $\mathcal{O}(\max_{i \in [n]} |\mathcal{A}_i|/\varepsilon^2)$ | Coarse Correlated Equilibrium |
| Algorithm 1 and 2 | $\mathcal{O}(\max_{i \in [n]} |\mathcal{A}_i|/\varepsilon^{11/4})$ | $\mathcal{O}(1/\varepsilon^{3/4})$ | Nash Equilibrium |

down decision making. Due to the high cost of acquiring information, agents are often forced to rely on outdated data to infer how their actions influence the group. This delay in feedback makes effective coordination and collaboration much more challenging. Therefore, it is crucial to design MARL algorithms that can reduce communication costs while still maintaining comparable performance.

In this work, following standard RL practice, we assume there exists a shared replay buffer that stores all data accessible to each agent, but agents can only get data in a communication round. The goal is to reduce the number of communication rounds, while still keeping decent convergence performance. A natural and widely used way to leverage old data for policy updates in empirical MARL algorithms is importance sampling, where gradients are estimated using samples collected under previous policies and are corrected by an importance weight. However, importance sampling suffers from large variance in the situation where the number of communication rounds is limited and the base policy is too outdated. To mitigate this issue, we provide a strategically modified importance sampling (IS) that can further decrease the number of communication rounds compared to previous works, while keeping comparable performance. The key technique of the modified IS is the *base policy prediction*. Unlike classical importance sampling, which keeps the base policy fixed for several future iterations and updates the new policy using gradient estimates from samples collected under the base policy, we predict the base policy by performing gradient updates but using only the old gradient. This approach allows us to get a smaller difference between the base and new policies, which in turn bounds the variance of the importance sampling estimates. As a result, it reduces the required communication rounds, since the smaller variance of importance sampling allows longer intervals and avoids frequent communication with the data buffer.

To be more specific, we make the following contributions:

- As a starting point, we study Potential Games (PGs), which is a fundamental subclass of MARL. We propose a novel algorithm that converges to a $\varepsilon$-Nash Equilibrium (NE) using $\mathcal{O}\left(\text{poly}(\max_i |\mathcal{A}_i|) \cdot \varepsilon^{-11/4}\right)$ samples, while requiring only $\mathcal{O}(\varepsilon^{-3/4})$ communication rounds.

- Furthermore, we extend our framework to general Markov Cooperative Games (MCGs), where the objective is to find a policy that can achieve agent-wise local optima of the total reward in MCG. We propose an algorithm that uses our Potential Game algorithm as an oracle to solve the equilibrium step-by-step, which can converge to a product policy that approximates the agent-wise local optima within $\mathcal{O}(\max_i |\mathcal{A}_i| \cdot \varepsilon^{-11/4})$ sample complexity and $\widetilde{\mathcal{O}}(1/\varepsilon^{3/4})$ communication cost. Table 1 provides comparisons with some previous representative works for both PGs and MCGs. Compared to (Liu et al., 2021), we avoid the dependence on exponential size of the joint action space in our result. Compared to (Song et al., 2021), our approach achieves better $\varepsilon$-dependence in both sample complexity and communication cost, and avoids dependence on size of action space in communication cost. Finally, compared to (Wang et al., 2023), which only converges to a Coarse Correlated Equilibrium (CCE), our results also have a significantly smaller communication round, with a slightly higher sample complexity.

- Empirically, we first test our algorithms in a simulated potential game and a congestion game. Then, we apply our base policy prediction idea to the MAPPO algorithm in more complicated environments. We show that our algorithms can reduce the communication cost while keeping comparable performance, while baseline algorithms with the same communication interval fail.

## 2 RELATED WORK

**Markov Games**  The Markov Game (MG) (Littman, 1994) is a widely used framework in multi-agent reinforcement learning. Several previous theoretical works (Liu et al., 2021; Tian et al., 2021; Liu et al., 2022) investigate learning NE and establish regret or sample complexity bounds that depend on the exponential action space. To break this exponential curse, subsequent works have proposed decentralized algorithms that focus on learning CCE or CE (Jin et al., 2021; Daskalakis et al., 2022; Zhan et al., 2022; Cui et al., 2023; Wang et al., 2023). However, these works still require access to other agents' actions at each time.

A line of research focuses on cooperative MARL, where agents aim to collaborate together. In the Markov Cooperative Game (MCG), each agent receives the same team reward, and the goal is to maximize this total reward through coordination. This formulation is widely adopted in empirical MARL studies (Yu et al., 2022; Lowe et al., 2017; Rashid et al., 2020; Sunehag et al., 2017). On the theoretical side, some works (Leonardos et al., 2021; Zhang et al., 2021; Ding et al., 2022) study the independent policy gradient ascent algorithm in MPG, providing algorithms for learning NE with polynomial sample complexity. (Sun et al., 2023) improves the previous convergence rate by introducing a suboptimality gap-dependent result.

**Importance Sampling in RL**  Many policy gradient approaches in RL use importance sampling to reuse the old data, which enables effective learning with less samples, like TRPO (Schulman et al., 2015), PPO (Schulman et al., 2017), and MARL algorithms like MAPPO (Yu et al., 2022) and DOP (Wang et al., 2020). Some works also consider reducing the variance of importance sampling in RL. Clipping is a common technique for variance reduction, as used in PPO (Schulman et al., 2017) and Retrace (Munos et al., 2016). The idea of combining importance sampling with a regression model to reduce the variance, known as AIPW (Rotnitzky et al., 1998) in the causal inference literature, was adapted to off-policy evaluation (OPE) in RL by (Jiang and Li, 2016; Thomas and Brunskill, 2016) named Doubly-Robust OPE. However, it has not been used in MARL to the best of our knowledge.

**Delayed Reward**  Our work is also similar to the study of delayed reward setting, in which the reward of the current step is given after several timesteps. There is a series of works studying delayed reward in Multi-Armed Bandit. Some works (Cesa-Bianchi et al., 2016) study the constant delay, while some other works (Gyorgy and Joulani, 2021; Zimmert and Seldin, 2020; Gael et al., 2020) consider adversarial delay. Beyond bandits, some papers also consider RL (Jin et al., 2022; Lancewicki et al., 2022; Zhan et al., 2025) and MARL (Zhang et al., 2023) with delayed reward. The key difference between these works and our work is that we allow agents to actively request rewards via communication, while they need to adapt to delays from some distribution.

## 3 PRELIMINARIES

Markov Games (MGs) can be represented by a tuple $(n, \mathcal{S}, \mathcal{A}, H, \{\mathbb{P}_h\}_{h \in [H]}, \{r_i\}_{i \in [n]})$, where $n \geq 2$ is the number of agents, $\mathcal{S}$ is the state space, $\mathcal{A} = \mathcal{A}_1 \times \mathcal{A}_2 \times \cdots \times \mathcal{A}_n$ is the joint action space for $n$ agents, $H$ is the horizon, $\mathbb{P}_h : \mathcal{S} \times \mathcal{A} \to \Delta(\mathcal{S})$ denotes the transition probability at step $h$, and $r_i : \mathcal{S} \times \mathcal{A} \mapsto [0, R_{\max}]$ is the reward function for agent $i$. In Markov Games, a policy for the agent $i$ is denoted as $\pi_i = \prod_{h=1}^{H} \pi_{i,h}$, where for each step $h$, the policy $\pi_{i,h} : \mathcal{S} \mapsto \Delta(\mathcal{A}_i)$ can be regarded as a distribution of the actions given the state. The joint policy is denoted by $\pi : \mathcal{S} \mapsto \Delta(\mathcal{A})$ where the randomness can be correlated. One particular class of policy is the product policy, where the randomness of each agent is independent and $\pi_h(s, \boldsymbol{a})$ can be rewritten as $\pi_h(s, \boldsymbol{a}) = \prod_{i=1}^{n} \pi_{i,h}(s, a_i)$. We define the state value function of agent $i$ under policy $\pi$ as $V_{i,h}^{\pi}(s) = \mathbb{E}_{\pi} \left[ \sum_{h'=h}^{H-1} r_i(s_{h'}, \boldsymbol{a}_{h'}) \mid s_h = s \right]$, where $(s_{h'}, \boldsymbol{a}_{h'})$ denotes the global state–action pair at time $h'$. We assume the initial state is fixed as $s_1$, and the total value function is $V_{i,1}^{\pi}(s_1)$.

The goal of a Markov game is to find an approximate equilibrium policy $\pi$ where no agent can benefit by deviating unilaterally. We define the equilibrium gap based on this concept.

**Definition 3.1** (Equilibrium Gap)**.** The equilibrium gap of a joint policy $\pi \in \Pi$ is defined by $\text{Gap}(\pi) = \max_{i \in [n]} \left( V_{i,1}^{\dagger \times \pi_{-i}}(s_1) - V_{i,1}^{\pi}(s_1) \right)$, where $V_{i,1}^{\dagger \times \pi_{-i}}(s_1)$ represents the value of best

response i.e., $V_{i,1}^{\dagger \times \pi_{-i}}(s_1) = \max_{\pi'_i \in \Pi_i} V_1^{\pi'_i \times \pi_{-i}}(s_1)$. In particular, a joint policy $\pi$ with an equilibrium gap less than $\varepsilon$ is called $\varepsilon$-CCE (**C**oarse **C**orrelated **E**quilibrium), and a product policy with an equilibrium gap less than $\varepsilon$ is called $\varepsilon$-NE (**N**ash **E**quilibrium).

A particularly important subclass is the Markov Cooperative Game (MCG), in which all agents share the same reward function $r_i \equiv r$ for all $i \in [n]$. Hence, we can simplify the value function as $V_h^\pi(s)$. In Markov Cooperative Game, the definition of equilibrium gap simplifies: $\pi$ has an equilibrium gap less than $\varepsilon$ if it is an approximate agent-wise local maximum of the common reward, i.e., $V_1^{\dagger \times \pi_{-i}}(s_1) \leq V_1^\pi(s_1) + \varepsilon$. That is, no single agent can unilaterally deviate and increase the team reward by more than $\varepsilon$. This agent-wise local maximum is important in cooperative MARL. In fact, an agent-wise local maximum is a point at which the total reward cannot be further increased through unilateral modifications, which implies a stable coordination system.

In this work, unlike classical games where agents receive rewards at every step, we assume rewards are stored in a buffer and can be accessed only during communication rounds. A communication round allows agents to exchange information but not interact with the environment, while sampling can only be done after the communication round. Formally, in a multi-agent potential game over $T$ rounds with communication at times $T_1, \cdots, T_r$, agents cannot observe others' actions or rewards between communication rounds, but at each round they can access the shared buffer and coordinate for future steps. We also call the number of communication rounds *the communication cost*.

## 4 A BASE CASE: POTENTIAL GAMES

To study the general MCG setting, we begin with a simpler base case known as Potential Games (PGs). In PGs, the rewards may differ across agents, and there exists a potential function $\phi$ that captures the value differences among them. As we will demonstrate in the next section, the algorithm developed for PGs serves as a fundamental building block and can be incorporated into the framework for solving general MCGs.

Formally, we assume that each agent $i$ has the approximated reward function $\hat{r}_i(\boldsymbol{a}) \in [0, R_{\max}]$ stored in the shared buffer, which is a bounded unbiased estimator of the true reward $r_i(\boldsymbol{a}) \in [0, R_{\max}]$. In a PG, there exists a potential function $\phi : \mathcal{A} \to [0, \phi_{\max}]$ that characterizes the unilateral deviation of all agents. We assume $M = \max\{R_{\max}, \phi_{\max}\} \geq 1$ in this section. Formally, denote $\phi(\pi) = \mathbb{E}_{\boldsymbol{a} \sim \pi}[\phi(\boldsymbol{a})]$, then the change in individual value resulting from a unilateral deviation can be captured by the potential function, i.e.

$$\mathbb{E}_{\boldsymbol{a} \sim \pi_i \times \pi_{-i}}[r_i(\boldsymbol{a})] - \mathbb{E}_{\boldsymbol{a} \sim \pi'_i \times \pi_{-i}}[r_i(\boldsymbol{a})] = \phi(\pi_i \times \pi_{-i}) - \phi(\pi'_i \times \pi_{-i}), \forall i \in [n]. \tag{1}$$

### 4.1 IMPORTANCE SAMPLING WITH BASE POLICY PREDICTION

In practical policy-gradient style algorithms like PPO or TRPO, the common solution for not collecting samples and rewards at each round is called Importance Sampling (IS). However, the variance of IS estimators can become large when the new and old policies diverge significantly, which presents a key challenge when we want to significantly reduce the communication cost, where the old policy will remain unchanged for a long time.

To control the variance of importance sampling, we propose *Base Policy Prediction*, where instead of fixing a single base policy and reusing it across multiple future iterations, we proactively predict a sequence of base policies in advance. During the communication round, we apply several gradient updates using the current gradient estimate to generate a number of predicted base policies. We then collect samples under all of these predicted base policies. In subsequent iterations, we use the corresponding predicted base policy as the reference distribution in importance sampling. This strategy ensures that the base and target policies remain closer in distribution, thereby reducing variance of IS estimator. This helps us to use old data for $\mathcal{O}(1/\varepsilon^{1/4}) > \Omega(1)$ rounds, and significantly reduces the number of communication rounds to get new data.

Our Algorithm 1 PG$(\mathcal{P}, T, \varepsilon)$ takes the potential game instance $\mathcal{P}$, total time $T$ and the target accuracy $\varepsilon$ as input. It is based on the Natural Policy Gradient (NPG) algorithm:

$$\hat{\pi}_i^{t+1}(a_i) \propto \hat{\pi}_i^t(a_i) \cdot \exp(\eta \ell_i(a_i, \hat{\pi}^t)), \tag{2}$$

---

**Algorithm 1** $\text{PG}(\mathcal{P}, T, \varepsilon)$

---

1: Initalize the policy $\pi^0 = \pi^{\text{base}}$.
2: **for** $t = 1, 2, \cdots, T$ **do**
3:     Calculate $\hat{\pi}_i^t(\mathcal{A}_i) \propto \hat{\pi}_i^{t-1}(\mathcal{A}_i) \cdot \exp(\eta \hat{\ell}_i(a_i, \hat{\pi}^{t-1}))$.    // Natural Policy Gradient
4:     **if** Changing Condition in Eq. 4 holds **then**
5:         Change the base policy $\hat{\pi}^t = \pi^{\text{base}}$.
6:         Compute $\overline{\pi}^{(t+t')}$ for $0 \le t' \le T$ by Eq. 3.
7:         Denote $\Pi_t = \{\overline{\pi}^{(t+t')} \mid 0 \le t' \le T\}$ as the set of distinct policies of $\overline{\pi}^{(t+t')}$.    // Base Policy Prediction
8:         **for** each $\pi \in \Pi^t$ **do**
9:             For $i \in [n]$ and $a_i \in \mathcal{A}_i$, sample $\boldsymbol{a}_1, \cdots, \boldsymbol{a}_N \sim \mathcal{A}_i \times \overline{\pi}_{-i}^{t+t'}$ to collect the $N = \Theta(1/\varepsilon^2)$ samples $\mathcal{D}_{\mathcal{A}_i, t+t'}^{(i)} = \{\hat{r}_i(\boldsymbol{a}_j)\}_{j \in [N], i \in [n]}$.    // Collect Samples for Base Policies
10:         **end for**
11:         Update $\mathcal{D}_{t+t'} = \bigcup_{i=1}^n \bigcup_{a_i \in \mathcal{A}_i} \mathcal{D}_{\mathcal{A}_i, \overline{\pi}^{(t+t')}}^{(i)}$ as the final dataset for each $0 \le t' \le T$.
12:         Start a communication round to share all the dataset $\mathcal{D}_{t+t'}$ for all $t'$ and the current policy $\hat{\pi}^t$ to all agents.    // Access the Data Buffer
13:     **end if**
14:     Reweight $\mathcal{D}_{\mathcal{A}_i, t}^{(i)} = \{(\mathcal{A}_i, \boldsymbol{a}_{-i}), \hat{r}_i(a_i, \boldsymbol{a}_{-i})\}_{n \in N}$ by $\hat{r}_i(a_i, \boldsymbol{a}_{-i}) \cdot \frac{\hat{\pi}^t(\boldsymbol{a}_{-i})}{\overline{\pi}^t(\boldsymbol{a}_{-i})}$. Get an estimate of $\hat{\ell}_i(a_i, \hat{\pi}^t) = \mathbb{E}_{\boldsymbol{a}_{-i} \sim \hat{\pi}_{-i}^t}[\hat{r}_i(a_i, \boldsymbol{a}_{-i})] = \mathbb{E}_{\mathcal{D}_{\mathcal{A}_i, t}^{(i)}}[\hat{r}_i(a_i, \boldsymbol{a}_{-i})]$ for each $a_i \in \mathcal{A}_i$.
15:     Calculate $g_{t-1} = \sum_{i=1}^n \{\mathbb{E}_{a_i \sim \hat{\pi}_i^t}[\hat{\ell}_i(\hat{\pi}_i^t, \hat{\pi}_{-i}^{t-1})] - \hat{\ell}_i(\hat{\pi}^{t-1})\}$.    // Estimate Nash Gap
16: **end for**
17: **Return** $\hat{\pi}^{k^*}$, where $k^* = \arg\min_{k \in [T-1]} g^k$.

---

where $\ell_i(a_i, \pi^t) = \mathbb{E}_{a_i \sim \pi_{-i}^t}[r_i(\boldsymbol{a})]$ is the marginalized reward function. The algorithm begins with a base policy $\pi^{\text{base}}$ and iteratively updates the joint policy over $T$ iterations. At each round $t$, the algorithm checks whether a predefined changing condition (Eq. 4) is satisfied. If so, the base policy is reset, and a collection of base policies $\overline{\pi}^{(t+t')}$ is constructed. For each base policy, data is collected by executing actions across all agents with sufficiently many samples. In the following paragraphs, we provide a detailed introduction to key components.

**Base Policy Prediction** The collection of base policies plays a central role in our algorithm. The natural policy gradient induces the update rule $\pi_i^{t+t'}(a_i) \propto \pi_i^t(a_i) \cdot \exp\left(\eta \sum_{j=t}^{t+t'-1} \ell_i(a_i, \pi^j)\right)$. We construct the base policy for $\pi_i^{t+t'}(\mathcal{A}_i)$ by

$$\tilde{\pi}_i^{t+t'}(a_i) \propto \pi_i^t(a_i) \cdot \exp\left(\eta t' \hat{\ell}_i(a_i, \pi^t)\right),$$

where $\hat{\ell}_i$ is the empirical estimate of $\ell_i$. This estimate is more accurate than old policy $\pi_i^t$ since $\hat{\ell}_i(a_i, \pi^t)$ can be regarded as an estimate of $\ell_i(a_i, \pi^{t+t'})$, if the difference between $\pi^t$ and $\pi^{t+t'}$ can be controlled. After getting $\tilde{\pi}_i^{t+t'}$, we further replace $\tilde{\pi}$ by $\overline{\pi}_i$ as follows. Define $\mathcal{A}_i^{t'} = \{a \in \mathcal{A}_i \mid \tilde{\pi}_i^{t+t'}(a) \le \varepsilon/|\mathcal{A}_i|\}$. Then, the $\overline{\pi}_i$ is defined by

$$\overline{\pi}_i^{t+t'}(a_i) = \begin{cases} \frac{\varepsilon}{|\mathcal{A}_i|}, & \text{if } a \in \mathcal{A}_i^{t'} \\ \left(1 - \frac{\varepsilon|\mathcal{A}_i^{t'}|}{|\mathcal{A}_i|}\right) \tilde{\pi}_i^{t+t'}(a_i), & \text{else} \end{cases}, \tag{3}$$

In fact, we redistribute the probability mass by mixing the previous base policy with the uniform distribution over $\mathcal{A}_i$. This modification introduces only an $O(\varepsilon)$ error for each policy, while substantially reducing the number of distinct policies in the collection $\{\pi^{t+t'}\}_{t' \in [T-t]}$. In fact, under Assumption 4.1, we can prove that for any $t$, the cardinality of the set $\Pi_t$ is bounded by

$$|\Pi_t| \le \sqrt{n} \log(\max_i |\mathcal{A}_i|/\varepsilon)/\Delta.$$

Such a reduction is crucial for improving the efficiency of sample collection, since we only need to collect samples for distinct policies. With the aggregated dataset, for the future steps, each agent

uses importance sampling to estimate its expected payoff $\hat{\ell}_i(a_i, \hat{\pi}^t)$ by the data sampled from $\overline{\pi}^t$. Since both the collected samples and the policies of all agents are shared during the most recent communication round, each agent $i$ has sufficient information to reconstruct the current joint policy $\hat{\pi}^t$ on its own, which avoids the extra communication at each round.

**Changing Condition** Another key component is the condition of communication, in which agents can get access to the global data $\mathcal{D}_{t+t'}$ and synchronize their policies $\hat{\pi}^t$. With this condition, we need to make sure a communication round is started whenever the base policy estimation becomes too outdated and the variance of importance sampling becomes too large. Note that from the base policy prediction, the variance of the importance sampling can be bounded if the difference of the marginalized reward $\sum_{j=0}^{t'-1} \ell_i(a_i, \hat{\pi}^{t+j}) - \ell_i(a_i, \hat{\pi}^t) = \mathcal{O}(1)$. Hence, our changing condition is designed to make sure that this difference will not exceed a threshold. To be more specific, our rule for starting a communication round is that there exists some $i$ and some actions $a_i \in \mathcal{A}_i$, such that

$$\underbrace{\left|\hat{\ell}_i(a_i, \hat{\pi}^{t+t'-1}) - \hat{\ell}_i(a_i, \hat{\pi}^t)\right| \geq \frac{16}{(n-1)t'\phi_{\max}\ln(1/n\varepsilon)}}_{(A)}, \text{ or } \underbrace{t' \geq \frac{1}{n^{1/4}\varepsilon^{1/4}\phi_{\max}\ln(1/n\varepsilon)}}_{(B)}. \quad (4)$$

Intuitively, the first condition ensures that the estimation error of the marginalized reward is small, which implies the variance of the importance sampling estimator remains bounded. The second condition prevents long periods without communication.

### 4.2 THEORETICAL RESULTS

First, we adopt the sub-optimality gap assumption in Sun et al. (2023).

**Assumption 4.1.** For any policy $\pi$, define $\mathcal{A}_{i,\pi}^* = \arg\max_{a \in \mathcal{A}_i} \ell_i(a, \pi)$ and $a_{i,\pi}^* \in \mathcal{A}_{i,\pi}^*$. Also, define $a_{i,\pi}^{**} = \arg\max_{a \in \mathcal{A}_i \setminus \mathcal{A}_{i,\pi}^*} \ell_i(a, \pi)$. Then, we define the suboptimality-gap for agent $i$ and $\pi$ as $\Delta_{i,\pi} = \ell_i(a_{i,\pi}^*, \pi) - \ell_i(a_{i,\pi}^{**}, \pi)$, and $c_{i,\pi} = \sum_{a_i \in \mathcal{A}_{i,\pi}^*} \pi_i(a_i)$. We assume that two constants $c > 0$ and $\Delta > 0$ such that

$$\min_{i \in [n], t \in [T]} c_{i,\hat{\pi}^t} \geq c, \qquad \min_{i \in [n], t \in [T]} \Delta_{i,\hat{\pi}^t} \geq \Delta.$$

Intuitively, $c > 0$ ensures that the algorithm does not converge to a point with a small gradient norm that is still far from the Nash equilibrium. Some work have shown that in such ill-conditioned cases, an exponential iteration complexity becomes unavoidable. $\Delta > 0$ indicates that there is a non-negligible gap between the gradients evaluated at the best and the second-best policies, which plays a crucial role in establishing the $\mathcal{O}(1/\varepsilon)$ convergence rate shown in Sun et al. (2023). Now we provide our results for the potential game.

**Theorem 4.2.** *Denote $M = \max\{\phi_{\max}, R_{\max}\} \geq 1$. Under Assumption 4.1, following Algorithm 1 with learning rate $\eta = \frac{1}{2nM}$, if $\varepsilon \leq \min\{1/n, 1/4\}$, with probability at least $1 - \delta$, policies $\{\hat{\pi}^t\}_{t \in [T]}$ satisfy that*

$$\frac{1}{T-1} \sum_{t=1}^{T-1} Gap(\hat{\pi}^t) \leq 8n\varepsilon \cdot M^2 \left(1 + \frac{1}{c\Delta}\right). \quad (5)$$

*when $T = 2/\varepsilon$. The total communication cost is bounded by $\mathcal{O}\left(\frac{n^{1/4}\phi_{\max}}{\varepsilon^{3/4}}\right)$, and the sample complexity is at most $\widetilde{\mathcal{O}}\left(\frac{nM^3(\sum_{i=1}^n |\mathcal{A}_i|)\log(1/\delta)}{\varepsilon^{11/4}\Delta}\right)$, where $\widetilde{\mathcal{O}}$ ignores some logarithm terms. Moreover, the output policy $\pi^{k^*}$ satisfies that $Gap(\hat{\pi}^{k^*}) \leq 28n\varepsilon \cdot M^2 \left(1 + \frac{1}{c\Delta}\right)^2$.*

Theorem 4.2 establishes that, given prior knowledge of the target accuracy $\varepsilon$, running the algorithm for $T = 1/\varepsilon$ iterations guarantees that the NE-gap converges to $\mathcal{O}(\varepsilon)$. The following corollary extends this result by showing that a convergence guarantee can still be obtained without prior knowledge of $\varepsilon$ and without a predefined time horizon.

**Corollary 4.3.** *Denote $\varepsilon_i = \frac{1}{2^{i-1}n}$ and $I_i = (2n \cdot (2^{i-1} - 1), 2n \cdot (2^i - 1)]$. For any Potential Game instance $\mathcal{P}$, during round $t \in I_i$, we execute $PG(\mathcal{P}, n \times 2^i, \varepsilon_i)$ to obtain the sequence of policies*

---

**Algorithm 2** MCG($T$)

---

1: **Initial**: $\pi^1$ to be the uniform policy $\pi^1_{i,h}(\cdot \mid s) = \text{Unif}(\mathcal{A}_i)$ for all $(i, s, h)$. $\mathcal{B}^0_h = \emptyset$.
2: **for** $t = 1, 2, \cdots, T$ **do**
3:     Execute $\pi^t$ to get $\{s_h\}_{h \in [H]}$. Collect $\mathcal{B}^t_h = \mathcal{B}^{t-1}_h \cup \{s_h\}$ in the shared buffer.
4:     **if** When $t - I_t = 2^a$ for some $a \in \mathbb{Z}$ **then**
5:         // Checking Changing Condition for Speicific Timestep
6:         Start the communication round to get $\mathcal{B}^t_h$ from the buffer. $T_h(s) = \sum_{s' \in \mathcal{B}^t_h} \mathbb{I}\{s' = s\}$.
7:         **if** $T^t_h(s) \geq 2T^{I_t}_h(s)$ **then**
8:             // Doubling Trick for Changing Condition
9:             Set exploration policy $\overline{\pi}^t \to \text{Unif}(\{\pi^\tau\}_{\tau \in [t]})$ and $\overline{V}^{t+1}_{H+1}(s) = 0$ for each $s \in \mathcal{S}$.
10:            **for** $h = H, \cdots, 1$ **do**
11:               Compute $\pi^{t+1}_h = \text{PG-Share}_h(\overline{\pi}^t, \overline{V}^{t+1}_{h+1}, \lfloor\sqrt{t}\rfloor + 1)$    // Potential Game Oracle
12:               Compute $\overline{V}^{t+1}_h = \text{V-Approx}_h(\overline{\pi}^t, \pi^{t+1}_h, \overline{V}^{t+1}_{h+1}, t)$.    // Backward Calculation
13:            **end for**
14:            Set $I_{t+1} = t$.
15:         **end if**
16:     **else**
17:         Set $\pi^{t+1}_h = \pi^t_h$. $I_{t+1} = I_t$.
18:     **end if**
19: **end for**
20: For each distinct policy $\pi \in \{\tilde{\pi}^t\}_{t \in [T]}$, collect $\mathcal{O}(H^2 \log(1/\delta)/\varepsilon^2)$ trajectories for each joint policy $a_i \times \pi_{-i}, i \in [n], a_i \in \mathcal{A}_i$. Start a communication round to get empirical estimates $V^\pi(s_1)$ and $\{V^{a_i \times \pi_{-i}}_1(s_1)\}$, denoted as $\hat{V}^\pi(s_1)$ and $\{\hat{V}^{a_i \times \pi_{-i}}_1(s_1)\}$.
21: Calculate $g(\pi) = \sum^n_{i=1} \max_{a_i \in \mathcal{A}_i} \hat{V}^{a_i \times \pi_{-i}}_1(s_1) - \hat{V}^\pi(s_1)$ for each different policy $\pi \in \{\tilde{\pi}^t\}_{t \in [T]}$. Denote $t^* = \min_{t \in [T]} g(\tilde{\pi}^t)$.    // Estimate Equilibrium Gap
22: **Return** $\tilde{\pi}^{t^*}$.

---

$\{\hat{\pi}^t\}_{t \in I_i}$. *We refer to this overall procedure as the **PG-Unknown**($\mathcal{P}$) protocol. Suppose the protocol is run for $T$ rounds, producing the sequence $\{\hat{\pi}^t\}_{t \in [T]}$. Define $s = \lfloor \log_2(T/2n) \rfloor$, then if the output of **PG-Unknown**($\mathcal{P}$) is defined as the product policy $\pi^{out} = PG(\mathcal{P}, n \times 2^s, \varepsilon_s)$, with probability at least $1 - \delta$, the output policy satisfies that*

$$Gap(\pi^{out}) \leq 28n\varepsilon_s \cdot M^3 \left(1 + \frac{1}{c\Delta}\right)^2 \leq \frac{280nM^3(1 + \frac{1}{c\Delta})^2}{T}$$

*with $\widetilde{\mathcal{O}}\left(\frac{nM^3(\sum^n_{i=1}|\mathcal{A}_i|)T^{11/4}}{\Delta}\right)$ sample complexity and $\widetilde{\mathcal{O}}\left(n^{1/4}\phi_{\max}T^{3/4}\right)$ communication cost.*

## 5 GENERAL MCG

Now we study the general MCG setting. We assume that the reward is bounded by $[0, 1]$, so the upper bound of the value function will be $H$. The key difficulty when applying the Potential Game oracle into this framework is that, the transition dynamics cannot be directly estimated through the transition kernel $\hat{\mathbb{P}}(s' \mid s, \boldsymbol{a})$ and dynamic programming, as the number of parameters in the transition kernel grows exponentially. Hence, we borrow the idea from V-learning (Wang et al., 2023), in which the algorithm learns the value function directly and apply the previous potential game oracle to update the policy.

To be more specific, at each iteration, we will sample from the policy $\pi^t$ to collect the data in $\mathcal{B}^t_h$, which is necessary for checking the trigger condition. Since checking the trigger condition needs communication for data sharing, we use a doubling trick (Line 4) to decide the timestep for checking the trigger condition. When the trigger condition is checked and satisfied, an exploration policy $\overline{\pi}^t$ is constructed by uniformly sampling from past policies. Then, starting from $\overline{V}^{t+1}_{H+1} = 0$, the algorithm iteratively applies two key subroutines: (i) the PG-Share subprocedure, which uses the potential game oracle to compute an updated policy $\pi^{t+1}_h$ based on the exploration policy and

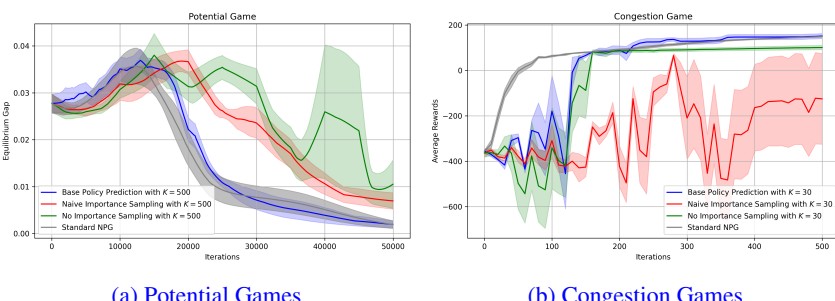

(a) Potential Games            (b) Congestion Games

Figure 1: Convergence Result for Different Algorithms.

the current value estimates at the step $h + 1$, and (ii) the V-Approx subprocedure, which updates the approximate value function $\overline{V}_h^{t+1}$ based on the exploration policy and the new policy. If the trigger condition is not met, the policy remains the same as the previous iteration. Finally, it estimates the equilibrium gap and outputs the policy with the minimum gap. The detailed introduction and analysis of the subprocedure PG-Share and V-Approx are provided in Appendix B. The following Theorem 5.1 guarantees that Algorithm 2 returns a $\mathcal{O}(\varepsilon)$ agent-wise local maximum.

**Theorem 5.1.** *Following Algorithm 2 with $T = 1/\varepsilon^2$, suppose Assumption 4.1 holds for all potential game instances we constructed in Algorithm 3, with probability at least $1 - \delta$, the outputted policy $\tilde{\pi}^{t^*}$ satisfied that*

$$Gap(\tilde{\pi}^{t^*}) \leq \widetilde{\mathcal{O}}\left(n^2 SH^4 \cdot \left(1 + \frac{1}{c\Delta}\right)^2\right) \cdot \varepsilon$$

*with $\widetilde{\mathcal{O}}\left(\frac{S^2 H^5 (\sum_{i=1}^n |\mathcal{A}_i|)}{\varepsilon^{11/4}\Delta}\right)$ sample complexity and $\widetilde{\mathcal{O}}\left(\frac{n^{1/4} S^2 H^3}{\varepsilon^{3/4}}\right)$ communication cost, which implies a $\widetilde{\mathcal{O}}(\varepsilon)$ agent-wise local maximum.*

## 6 EXPERIMENTS

In this section, we test our base policy prediction algorithms in both simulated games and more complicated environments. Since all previous theoretical works in Table 1 have no empirical results, we compare our base policy prediction algorithm with baseline algorithms (standard NPG or MAPPO) with the same communication interval.

**Potential Games** We follow the experimental setup in Sun et al. (2023), but modify the action space of three agents to be of size 10. More details are provided in Appendix C. We set the number of episodes to $N = 5000$. Then, let parameter $K$ indicates that a communication round is initiated after every $K$ episodes. Thus, there are in total $\lceil 5000/K \rceil$ communication rounds. The baseline algorithms are NPG without importance sampling and NPG with naive importance sampling, where the old policy is used as the base policy throughout the 500 episodes between communication rounds. Then, we implement our Base Policy Prediction (BPP) algorithm, in which the base policy is updated using the old gradient every 100 episodes. To provide the empirical evidence that our BPP mechanism successfully reduce the number of communication rounds without destroying the performance, we also add standard NPG that communicates and receives the reward for each episode. Figure 1a shows that with $K = 500$ and $5000/500 = 10$ communication rounds, our algorithm has the best performance across other algorithms with the same number of communication rounds, and performs as good as standard NPG, which communicates at each episode..

**Congestion Games** We adopt the congestion game setting from (Sun et al., 2023). The environment details are provided in Appendix C. We set the communication interval as 30, and we precompute the base policy for every 5 rounds in our algorithm. The baseline algorithms are the same as above. The reward curves are shown in Figure 1b, which shows that our approach achieves the highest rewards, and the naive algorithm without importance sampling converges to a suboptimal solution due to biased gradient estimates. Moreover, when the communication interval is large, the base policy in the naive importance sampling method becomes overly outdated, leading to instability and failure to converge.

**Experiments on MAPPO** We also evaluate our base policy prediction algorithm on two Multi-Agent Particle Environments (MPE) (Lowe et al., 2017), *Spread* and *Reference*, and three SMAC environments *1c3s5z*, *MMM* and *3s_vs_3z*. In the SMAC environment, the communication interval is required to be large only after the win rate exceeds a specified threshold (set to 30% in our experiments), in order to avoid the cold-start issue. We choose MAPPO (Yu et al., 2022) as our baseline algorithm. In MAPPO, agents receive new data every 10 episodes. We extend this by introducing a communication interval $I$, where agents collect data only once every $10 \times I$ gradient updates. At each communication round, $I$ base policies are predicted, and each base policy is used for 10 updates. In the first experiment with Figure 2, we show that a larger $I$ slows convergence but achieves comparable final performance, even with communication intervals 10 times longer. This demonstrates that the base policy prediction algorithm enables effective learning even when the communication interval is much larger than usual. The second experiment (rightmost two figures in Figure 2 shows that when the communication interval in naive MAPPO is scaled to $10 \times 10 = 100$, the algorithm fails to converge. These findings indicate that our method can both preserve the convergence guarantees of MAPPO and significantly reduce the communication cost.

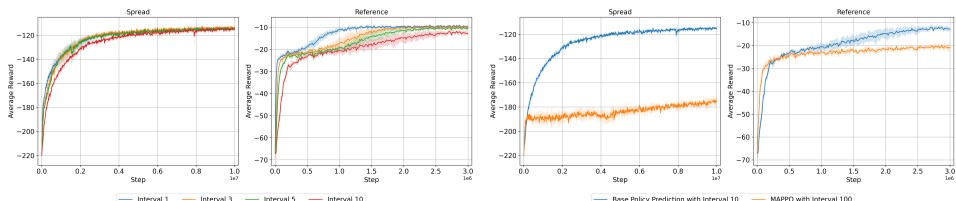

Figure 2: The Left two figures show the comparisons of base policy prediction under different communication intervals. The right two figures show the comparisons between MAPPO and base policy prediction.

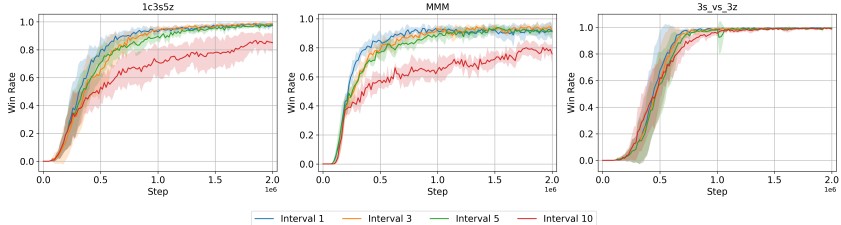

(a) Convergence results for Base Policy Prediction with different intervals

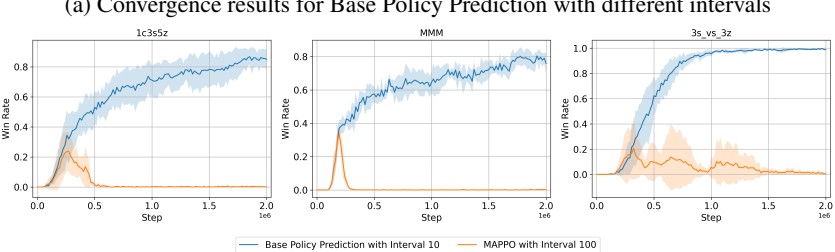

(b) Comparison with Naive MAPPO

Figure 3: Convergence results for SMAC environment

## 7 CONCLUSION

In this work, we study multi-agent RL under communication constraints, with the goal of reducing communication rounds during learning. We introduce a base policy prediction technique that pre-computes a series of base policies for future importance sampling. Theoretically, we show that NPG with importance sampling and base policy prediction improves prior results in both communication

efficiency and sample complexity. We further extend our potential game algorithm to general MCGs and design an algorithm that returns a product policy approximating agent-wise local optima. Empirically, our method enables effective learning while substantially reducing communication rounds compared to baseline approaches.

## ETHICS

This paper studies multi-agent reinforcement learning with communication constraints. We do not identify any ethical concerns associated with this research.

## REPRODUCIBILITY STATEMENT

The supplementary material contains the codebase for this paper.

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

# Appendix

## A   ANALYSIS OF PG ALGORITHMS

### A.1   PROOF OF THEOREM 4.2

We first re-study the natural policy gradient descent. Suppose at round $t$, we can get the reward function $r$ and the policy $\pi^t$, we can define the marginalized loss function as

$$\ell_i(a, \pi^t) = \mathbb{E}_{\boldsymbol{a}_{-i} \sim \pi_{-i}^t}[r_i(\boldsymbol{a})] \geq 0.$$

Then, the natural policy gradient algorithm has the following update:

$$\pi_i^{t+1} \propto \pi_i^t \cdot \exp(\eta \ell_i(a, \pi^t)).$$

Now we analyze the interval between communication rounds. If the policy at the last communication round is $\pi^{\text{base}}$, without loss of generality, we can assume that the last communication round is round 0 such that $\pi_0 = \pi^{\text{base}}$. Then, performing natural policy gradient update for $t$ rounds yields

$$\pi_i^{t+1}(a) \propto \pi_i^{\text{base}}(a) \cdot \exp(\eta \sum_{j=0}^{t} \ell_i(a, \pi^j)). \tag{6}$$

We now use importance sampling to get the approximation of $\pi^t(a)$.

We denote the approximated one of the $\pi^t$ as $\hat{\pi}^t$. Now we analyze the interval between communication rounds. We suppose that we can share the policy and get the policy $\pi^0$, we can denote $\hat{\pi}^0 = \pi^0$. We first estimate the marginalized loss function as

$$\hat{\ell}_i(a_i, \hat{\pi}^0) = \frac{1}{|\mathcal{D}_{\mathcal{A}_i,0}^{(i)}|} \sum_{r_i(\boldsymbol{a}) \in \mathcal{D}_{\mathcal{A}_i,0}^{(i)}} r_i(\boldsymbol{a}) \approx \mathbb{E}_{\boldsymbol{a} \sim \mathcal{A}_i \times \hat{\pi}_{-i}^0}[r_i(\boldsymbol{a})]$$

for each action $a_i \in \mathcal{A}_i$ using samples from $\hat{\pi}^0$. Then, to estimate the update Eq.6, we calculate the following term:

$$\tilde{\pi}_i^{t+1}(a) \propto \hat{\pi}_i^{\text{base}}(a) \cdot \exp\left(\eta \sum_{j=0}^{t} \hat{\ell}_i(a, \hat{\pi}^0)\right) = \hat{\pi}_i^{\text{base}}(a) \cdot \exp(\eta(t+1)\hat{\ell}_i(a, \hat{\pi}^0)), \tag{7}$$

and the dataset $\mathcal{D}_t = \bigcup_{i=1}^{n} \bigcup_{a_i \in \mathcal{A}_i} \mathcal{D}_{\mathcal{A}_i,t}^{(i)}$ is calculated by executing the action following the distribution $a_i \times \tilde{\pi}_{-i}^t(\boldsymbol{a}_{-i})$ for each agent $i$ and $a_i \in \mathcal{A}_i$.

Then, at the time $t$, suppose we have derived $\hat{\pi}^1, \cdots, \hat{\pi}^t$ and $\{\hat{\ell}(a, \hat{\pi}^j)\}_{j \leq t}$ at this interval without communication round, we let

$$\hat{\pi}_i^{t+1}(a) \propto \pi_i^{\text{base}}(a) \cdot \exp(\eta \sum_{j=0}^{t} \hat{\ell}_i(a, \hat{\pi}^j)),$$

and then $\ell_i(a, \hat{\pi}_h^t)$ can be estimated using importance sampling and shared data. To be more specific, we use

$$\hat{\ell}_i(a_i, \pi^t) = \mathbb{E}_{\boldsymbol{a}_{-i} \sim \hat{\pi}_{-i}^t}[r_i(a_i, \boldsymbol{a}_{-i})]$$

$$= \mathbb{E}_{\boldsymbol{a}_{-i} \sim \tilde{\pi}_{-i}^t}\left[\frac{\hat{\pi}_{-i}^t(\boldsymbol{a}_{-i})}{\overline{\pi}_{-i}^t(\boldsymbol{a}_{-i})} r_i(a_i, \boldsymbol{a}_{-i})\right]$$

$$= \mathbb{E}_{\boldsymbol{a}_{-i} \sim \mathcal{D}^t}\left[\frac{\hat{\pi}_{-i}^t(\boldsymbol{a}_{-i})}{\overline{\pi}_{-i}^t(\boldsymbol{a}_{-i})} \hat{r}_i(a_i, \boldsymbol{a}_{-i})\right].$$

as the estimated loss function. Now we analyze the communication rounds and the iteration complexity. First, we want to show that for any $t \geq 0$, we have

$$|\hat{\ell}_i(a, \hat{\pi}^t) - \ell_i(a, \hat{\pi}^t)| \leq \varepsilon. \tag{8}$$

For $t = 0$, which implies that the interval ends at time $t$, $\mathcal{D}^{(i)}_{\mathcal{A}_i,0}$ is collected from $\mathcal{A}_i \times \hat{\pi}^0$ and $\hat{r}_i$ is a unbiased and bounded estimate of $r_i$. Hence, we have

$$|\hat{\ell}_i(a, \hat{\pi}^0) - \ell_i(a, \hat{\pi}^0)| = \left| \frac{1}{|\mathcal{D}^{(i)}_{\mathcal{A}_i,0}|} \sum_{\hat{r}_i(\boldsymbol{a}) \in \mathcal{D}^{(i)}_{\mathcal{A}_i,0}} \hat{r}_i(\boldsymbol{a}) - \mathbb{E}_{\boldsymbol{a} \sim \mathcal{A}_i \times \hat{\pi}^0_{-i}}[r_i(\boldsymbol{a})] \right| \leq \frac{R_{\max}}{\sqrt{2N}} \sqrt{\log(2/\delta)} \leq \varepsilon.$$

The final inequality follows from Hoeffding's concentration inequality and the fact that $N \geq \frac{R_{\max}^2 \log(2/\delta)}{\varepsilon^2}$.

For each $t \geq 1$, to apply Hoeffding's inequality, since the data point is bounded as $R_{\max}$, we need to verify that the reweighted estimator remains bounded as a constant. Hence, we need to check the likelihood ratio between the base policy and the current policy is bounded by a constant. Now we analyze the likelihood ratio of the importance sampling, i.e.,

$$\frac{\hat{\pi}^t_{-i}(\boldsymbol{a}_{-i})}{\overline{\pi}^t_{-i}(\boldsymbol{a}_{-i})} = \frac{\hat{\pi}^t_{-i}(\boldsymbol{a}_{-i})}{\tilde{\pi}^t_{-i}(\boldsymbol{a}_{-i})} \cdot \frac{\tilde{\pi}^t_{-i}(\boldsymbol{a}_{-i})}{\overline{\pi}^t_{-i}(\boldsymbol{a}_{-i})} = \prod_{i' \neq i} \frac{\hat{\pi}^t_{i'}(a_{i'})}{\tilde{\pi}^t_{i'}(a_{i'})} \cdot \frac{\tilde{\pi}^t_{i'}(a_{i'})}{\overline{\pi}^t_{i'}(a_{i'})}.$$

Hence, we focus on controlloing the ratio $\frac{\hat{\pi}^t_i(a_i)}{\tilde{\pi}^t_i(a_i)}$ for each $i \in [n]$. Suppose Eq 8 holds for any $t' \leq t - 1$, then for the time $t$, by the Eq.2 and Eq.7, for each $i \in [n]$, we have

$$\frac{\hat{\pi}^t_i(a_i)}{\tilde{\pi}^t_i(a_i)}$$

$$= \exp\left( \eta \sum_{j=0}^{t-1} \hat{\ell}_i(a_i, \hat{\pi}^j) - \eta t \hat{\ell}_i(a_i, \hat{\pi}^0) \right) \cdot \frac{\sum_{a \in \mathcal{A}_i} \pi^{\text{base}}(a) \cdot \exp(\eta \sum_{j=0}^{t-1} \hat{\ell}_i(a_i, \hat{\pi}^0))}{\sum_{a \in \mathcal{A}_i} \pi^{\text{base}}(a) \cdot \exp(\eta \sum_{j=0}^{t-1} \hat{\ell}_i(a_i, \hat{\pi}^j))}$$

$$\leq \exp\left( \max_{a_i \in \mathcal{A}_i} \left| \eta \sum_{j=0}^{t-1} \hat{\ell}_i(a_i, \hat{\pi}^j) - t \hat{\ell}_i(a_i, \hat{\pi}^0) \right| \right) \cdot \frac{\sum_{a \in \mathcal{A}_i} \pi^{\text{base}}(a) \cdot \exp(\sum_{j=0}^{t-1} \hat{\ell}_i(a_i, \hat{\pi}^0))}{\sum_{a \in \mathcal{A}_i} \pi^{\text{base}}(a) \cdot \exp(\eta \sum_{j=0}^{t-1} \hat{\ell}_i(a_i, \hat{\pi}^j))}$$

$$\leq \exp\left( 2 \max_{a_i \in \mathcal{A}_i} \left| \sum_{j=0}^{t-1} \hat{\ell}_i(a_i, \hat{\pi}^j) - t \hat{\ell}_i(a_i, \hat{\pi}^0) \right| \right). \tag{9}$$

The first inequality uses the fact that $\eta = 1/8n \leq 1$. The last inequality uses the fact that

$$\sum_{a \in \mathcal{A}_i} \pi^{\text{base}}(a) \exp(\sum_{j=0}^{t-1} \hat{\ell}_i(a_i, \hat{\pi}^0))$$

$$\leq \max_{a_i \in \mathcal{A}_i} \exp\left( \left| \sum_{j=0}^{t-1} \hat{\ell}_i(a_i, \hat{\pi}^j) - t \hat{\ell}_i(a_i, \hat{\pi}^0) \right| \right) \cdot \sum_{a \in \mathcal{A}_i} \pi^{\text{base}}(a) \exp(\sum_{j=0}^{t-1} \hat{\ell}_i(a_i, \hat{\pi}^j))$$

since each $\pi^{\text{base}}(a) \geq 0$.

Now we consider the term $\frac{\tilde{\pi}^t_i(a_i)}{\overline{\pi}^t_i(a_i)}$. By definition, we will have

$$\frac{\tilde{\pi}^t_i(a_i)}{\overline{\pi}^t_i(a_i)} \leq \frac{1}{1 - \varepsilon}.$$

This inequality is because $\tilde{\pi}^t_i(a_i) > \overline{\pi}^t_i(a_i)$ implies that $a_i \notin \mathcal{A}^t_i$. Hence, $\frac{\tilde{\pi}^t_i(a_i)}{\overline{\pi}^t_i(a_i)} = 1 - \frac{\varepsilon |\mathcal{A}^t_i|}{|\mathcal{A}_i|} \geq 1 - \varepsilon$.

Hence, if the error $\varepsilon \leq 1/n$, the variance is bounded by

$$\frac{\hat{\pi}^t_{-i}(\boldsymbol{a}_{-i})}{\overline{\pi}^t_{-i}(\boldsymbol{a}_{-i})} \leq \frac{1}{(1 - \varepsilon)^n} \prod_{k \neq i} \max_{a_i \in \mathcal{A}_i} \exp(\sum_{j=0}^{t-1} \hat{\ell}_i(a_i, \hat{\pi}^j) - t \hat{\ell}_i(a_i, \hat{\pi}^0))$$

$$\leq \frac{1}{(1 - \frac{1}{n})^n} \cdot \exp\left((n-1) \cdot \max_i \max_{a_i \in \mathcal{A}_i} \left(\sum_{j=0}^{t-1} \hat{\ell}_i(a_i, \hat{\pi}^j) - t\hat{\ell}_i(a_i, \hat{\pi}^0)\right)\right)$$

$$\leq 2e \cdot \exp\left((n-1) \cdot \max_i \max_{a_i \in \mathcal{A}_i} \left(\sum_{j=0}^{t-1} \hat{\ell}_i(a_i, \hat{\pi}^j) - t\hat{\ell}_i(a_i, \hat{\pi}^0)\right)\right). \tag{10}$$

We now turn to controlling the variance bound. The key idea is that whenever the upper bound in Eq. 10 exceeds a prescribed threshold, or when the number of rounds since the last communication becomes too large, all agents simultaneously initiate a new communication round. To be more specific, our rule for starting a communication round is that there exists some $i$ and some actions $a_i \in \mathcal{A}_i$, such that

$$\underbrace{\left|\hat{\ell}_i(a_i, \hat{\pi}^{t-1}) - \hat{\ell}_i(a_i, \hat{\pi}^0)\right| \geq \frac{16}{(n-1)t\phi_{\max}\ln(1/n\varepsilon)}}_{(A)}, \quad \text{or } \underbrace{t \geq \frac{1}{n^{1/4}\varepsilon^{1/4}\phi_{\max}\ln(1/n\varepsilon)}}_{(B)}. \tag{11}$$

Intuitively, the first condition ensures that the variance of the importance sampling estimator remains uniformly bounded, thereby guaranteeing the reliability of subsequent updates. The second condition prevents long time without communication, which could otherwise lead to out-dated policy for importance sampling and divergence across agents.

Since the requirement (A) and (B) are not satisfied during the interval, we know that for all $j \in [t-1]$, $i \in [n]$ and $a_i \in \mathcal{A}_i$, the changing condition does not hold, it implies that the variance is bounded by a constant, i.e.,

$$\frac{\hat{\pi}_{-i}^t(\boldsymbol{a}_{-i})}{\overline{\pi}_{-i}(\boldsymbol{a}_{-i})} \leq 2e \cdot \exp\left((n-1) \cdot \sum_{j=0}^{t-1} \frac{16}{(n-1)(j+1)\phi_{\max}\ln(1/n\varepsilon)}\right)$$

$$\leq 2e \cdot \exp\left(\frac{16(1 + \ln t)}{\phi_{\max}\ln(1/n\varepsilon)}\right)$$

$$\leq 2\exp(17).$$

The second inequality uses the fact that $\sum_{j=1}^t \frac{1}{t} \leq 1 + \ln(t)$ and the upper bound of the $t$ in (B).

Hence, by getting $N \geq \widetilde{\mathcal{O}}\left(\frac{M^2 \log(1/\delta)}{\epsilon^2}\right)$ samples for each agent $i$ and each action $a_i \in \mathcal{A}_i$, we can get

$$|\hat{\ell}_i(a_i, \hat{\pi}^t) - \ell_i(a_i, \hat{\pi}^t)| \leq \varepsilon/2, \forall i \in [n], a_i \in \mathcal{A}_i \tag{12}$$

for any timestep $t$. Now we analyze when the interval ends, which implies that either (A) or (B) will hold. We assume that at time step $t$, the update rule is satisfied and interval ends.

**Situation 1: (A) holds**  If at time step $t$, the update rule (A) is satisfied, then there exists one $i$ and $a_i \in \mathcal{A}_i$ such that

$$\frac{16}{(n-1)t\phi_{\max}\ln(1/\sqrt{n\varepsilon})} \leq \left|\hat{\ell}_i(a_i, \hat{\pi}^{t-1}) - \hat{\ell}_i(a_i, \hat{\pi}^0)\right| \leq \varepsilon + \left|\ell_i(a_i, \hat{\pi}^{t-1}) - \ell_i(a_i, \hat{\pi}^0)\right|.$$

Now denote $\pi_{-i}^{a:b} = \prod_{k \neq i, a \leq k \leq b} \pi_k$. Then, for any $\pi, \pi'$, we have

$$\ell_i(a_i, \pi) - \ell_i(a_i, \pi') = \mathbb{E}_{\boldsymbol{a} \sim a_i \times \pi_{-i}}[r_i(\boldsymbol{a})] - \mathbb{E}_{\boldsymbol{a} \sim a_i \times \pi'_{-i}}[r_i(\boldsymbol{a})]$$

$$= \sum_{k \neq i} \mathbb{E}_{\boldsymbol{a} \sim a_i \times \pi_{-i}^{1:k-1} \times (\pi'_{-i})^{k:n}}[r_i(\boldsymbol{a})] - \mathbb{E}_{\boldsymbol{a} \sim a_i \times \pi_{-i}^{1:k-1} \times (\pi'_{-i})^{k:n-1}}[r_i(\boldsymbol{a})]$$

$$= \sum_{k \neq i} \phi(a_i \times \pi_{-i}^{1:k-1} \times (\pi'_{-i})^{k:n}) - \phi(a_i \times \pi_{-i}^{1:k} \times (\pi'_{-i})^{k+1:n-1})$$

$$= \phi(a_i \times \pi_{-i}) - \phi(a_i \times \pi'_{-i}),$$

we can get

$$\varepsilon + \left| \ell_i(a_i, \hat{\pi}^{t-1}) - \ell_i(a_i, \hat{\pi}^0) \right| = \varepsilon + \left| \phi(a_i \times \hat{\pi}_{-i}^{t-1}) - \phi(a_i \times \hat{\pi}_{-i}^0) \right|,$$

which implies that

$$\left| \phi(a_i \times \hat{\pi}_{-i}^{t-1}) - \phi(a_i \times \hat{\pi}_{-i}^0) \right| \geq \frac{16}{\;(n-1)t\phi_{\max}\ln(1/n\varepsilon)} - \varepsilon \geq \frac{16}{\sqrt{2}(n-1)t\phi_{\max}\ln(1/n\varepsilon)}, \tag{13}$$

The last inequality is because $t \leq \frac{1}{n^{1/4}\varepsilon^{1/4}\phi_{\max}\ln(1/n\varepsilon)}$. Then it implies that

$$\varepsilon \leq \frac{1}{nt^4\phi_{\max}^4\ln(1/n\varepsilon)^4} \leq \frac{16}{2\sqrt{2}(n-1)t\phi_{\max}\ln(1/n\varepsilon)}.$$

Now we denote $\pi_i^t(a_i) \propto \hat{\pi}_i^{t-1}(a_i) \cdot \exp(\eta\ell_i(a_i, \hat{\pi}^{t-1}))$ is the true policy with the gradient update, since $\hat{\pi}_i^t(a_i) \propto \hat{\pi}_i^{t-1}(a_i) \cdot \exp(\eta\hat{\ell}_i(a_i, \hat{\pi}^{t-1}))$, by the estimate error bound Eq 12 we can use the following lemma to bound the TV distance between two policies.

**Lemma A.1.** *Suppose $\pi_1 \propto \pi \cdot \exp(\eta f_1(a))$, $\pi_2 \propto \pi \cdot \exp(\eta f_2(a))$ with $|f_1(a) - f_2(a)| \leq \varepsilon/2$ for any action $a$. Then, the TV distance is bounded by*

$$TV(\pi_1, \pi_2) \leq \eta\varepsilon.$$

*Proof.*

$$\begin{aligned} TV(\pi_1, \pi_2) &= \sum_{a \in \mathcal{A}} \left( \frac{\pi(a) \cdot \exp(\eta f_1(a))}{\sum_{a' \in \mathcal{A}} \pi(a') \cdot \exp(\eta f_1(a'))} - \frac{\pi(a) \cdot \exp(\eta f_2(a))}{\sum_{a' \in \mathcal{A}} \pi(a') \cdot \exp(\eta f_2(a'))} \right) \\ &\leq \sum_{a \in \mathcal{A}} \frac{\pi(a) \cdot \exp(\eta f_1(a))}{\sum_{a' \in \mathcal{A}} \pi(a') \cdot \exp(\eta f_1(a'))} \cdot \left( 1 - \frac{\exp(-\eta\varepsilon/2)}{\exp(\eta\varepsilon/2)} \right) \\ &\leq 1 - e^{-\eta\varepsilon} \leq \eta\varepsilon. \end{aligned}$$

$\square$

Hence, by the subadditivity of the TV distance, we have

$$TV(\hat{\pi}^t, \pi^t) \leq \sum_{i=1}^{n} TV(\hat{\pi}_i^t, \pi_i^t) \leq \eta n\varepsilon \leq \varepsilon/2\phi_{\max}.$$

The last inequality uses the fact that $\eta = \frac{1}{2nM} \leq \frac{1}{2n\phi_{\max}}$. (Recall that $M = \max\{\phi_{\max}, R_{\max}\}$. Then we can derive

$$\begin{aligned} \left| \phi(a_i \times \hat{\pi}_{-i}^{t-1}) - \phi(a_i \times \hat{\pi}_{-i}^0) \right| &\leq \sum_{j=0}^{t-2} \left| \phi(a_i \times \hat{\pi}_{-i}^{j+1}) - \phi(a_i \times \hat{\pi}_{-i}^j) \right| \\ &\leq t \cdot (\varepsilon/2\phi_{\max}) \cdot \phi_{\max} + \sum_{j=0}^{t-2} \left| \phi(a_i \times \pi_{-i}^{j+1}) - \phi(a_i \times \hat{\pi}_{-i}^j) \right| \\ &\leq t\varepsilon/2 + \sum_{j=0}^{t-2} \phi_{\max} \cdot TV(a_i \times \pi_{-i}^{j+1}, a_i \times \hat{\pi}_{-i}^j) \\ &\leq t\varepsilon/2 + \sum_{j=0}^{t-2} \frac{\phi_{\max}}{\sqrt{2}} \cdot \sqrt{\mathrm{KL}(a_i \times \pi_{-i}^{j+1} \| a_i \times \hat{\pi}_{-i}^j)}. \end{aligned} \tag{14}$$

The last inequality uses the Pinsker's inequality that $TV(P, Q)^2 \leq \frac{1}{2}\mathrm{KL}(P\|Q)$. Now we know that

$$\mathrm{KL}(a_i \times \pi_{-i}^{j+1} \| a_i \times \hat{\pi}_{-i}^j) = \sum_{k \neq i} \mathrm{KL}(\pi_k^{j+1} \| \hat{\pi}_k^j) \leq \mathrm{KL}(\pi^{j+1} \| \hat{\pi}^j).$$

Hence, by Eq 13 and Eq 14, we have

$$\frac{16}{(n-1)t\phi_{\max}^2 \ln(1/n\varepsilon)} \leq t\varepsilon/2 + \sum_{j=0}^{t-2} \sqrt{\mathrm{KL}(\pi^{j+1}\|\hat{\pi}^j)}, \tag{15}$$

which implies that

$$\sum_{j=0}^{t-2} \sqrt{\mathrm{KL}(\pi^{j+1}\|\hat{\pi}^j)} \geq \frac{16}{\sqrt{2}(n-1)t\phi_{\max}^2 \ln(1/n\varepsilon)}. \tag{16}$$

for $\varepsilon \leq \frac{1}{n^2 t^4 \phi_{\max}^4 \ln^4(1/n\varepsilon)}$. Now we consider the KL divergence term. We first introduce the potential difference lemma (Sun et al., 2023), which can help us to link the KL divergence with the potential gap.

**Lemma A.2.** *[(Sun et al., 2023)] For $\pi_i^{j+1} \propto \hat{\pi}_i^j(a) \cdot \exp(\eta \ell_i(a, \hat{\pi}^j))$ with $\eta = \frac{1}{2nM}$, we have*

$$\phi(\pi^{j+1}) - \phi(\hat{\pi}^j) \geq (1 - R_{\max}\sqrt{n}\eta) \sum_{i=1}^{n} \langle \pi_i^{j+1} - \hat{\pi}_i^j, \ell(\cdot, \hat{\pi}^j) \rangle$$

$$= \frac{1 - R_{\max}\sqrt{n}\eta}{\eta} \cdot KL(\pi_i^{j+1}\|\hat{\pi}_i^j)$$

$$\geq n\phi_{\max} KL(\pi_i^{j+1}\|\hat{\pi}_i^j) \geq 0.$$

*Proof.* The proof of the first inequality is exactly the same as Lemma 3.1 in (Sun et al., 2019). The second line uses the definition of the KL divergence, and the last line uses the fact that $1 - R_{\max}\sqrt{n}\eta \geq 1/2$ and $\eta \leq \frac{1}{2n\phi_{\max}}$. $\qquad\square \qquad\qquad\qquad\square$

Hence, by Eq. 8 and Lemma A.1, we know that

$$TV(\hat{\pi}^{j+1}, \pi^{j+1}) \leq \sum_{i=1}^{n} TV(\hat{\pi}_i^{j+1}, \pi_i^{j+1}) \leq \eta n\varepsilon. \tag{17}$$

The last inequality uses the fact that $1 - \eta\varepsilon \geq 1 - \frac{\varepsilon}{8n} \geq 1/2$. Then, we can get

$$\phi(\hat{\pi}^{j+1}) - \phi(\hat{\pi}^j) \geq \phi(\hat{\pi}^{j+1}) - \phi(\pi^{j+1}) \geq 0 - \eta\phi_{\max}n\varepsilon \geq -\eta\phi_{\max}n\varepsilon \geq -\varepsilon/2.$$

Combining Lemma A.2 and Eq 15, we have

$$\frac{16}{\sqrt{2}(n-1)t\phi_{\max}^2 \ln(1/n\varepsilon)} \leq \sum_{j=0}^{t-2} \sqrt{\frac{\phi(\pi^{j+1}) - \phi(\hat{\pi}^j)}{n\phi_{\max}}} \leq \sqrt{2t} \cdot \sqrt{\sum_{j=0}^{t-2} \left(\frac{\phi(\pi^{j+1}) - \phi(\hat{\pi}^j)}{n\phi_{\max}}\right)}$$

$$\leq \sqrt{2t} \cdot \sqrt{\frac{t\varepsilon}{2n\phi_{\max}} + \sum_{j=0}^{t-2} \frac{\phi(\hat{\pi}^{j+1}) - \phi(\hat{\pi}^j)}{n\phi_{\max}}}$$

$$\leq \sqrt{2t} \cdot \sqrt{\frac{t\varepsilon}{2n\phi_{\max}} + \frac{\phi(\hat{\pi}^{t-1}) - \phi(\hat{\pi}^0)}{n\phi_{\max}}},$$

which implies that

$$\frac{256}{4(n-1)^2 t^3 \phi_{\max}^4 \ln(1/n\varepsilon)^2} - \frac{t\varepsilon}{2n\phi_{\max}} \leq \frac{\phi(\hat{\pi}^{t-1}) - \phi(\hat{\pi}_0)}{n\phi_{\max}}.$$

By some simple algebra and $R_{\max} \geq 1$, we can get

$$\phi(\hat{\pi}^{t-1}) - \phi(\hat{\pi}_0) \geq \frac{256 R_{\max}}{4nt^3 \phi_{\max}^3 \ln(1/n\varepsilon)^2} - \frac{t\varepsilon}{2} \geq \frac{256}{6nt^3 \phi_{\max}^3 \ln(1/n\varepsilon)^2}$$

The last inequality uses the fact that $t \leq \frac{1}{n^{1/4}\varepsilon^{1/4}\phi_{\max}\ln(1/n\varepsilon)}$, which implies that

$$\frac{t\varepsilon}{2} \leq \frac{256}{12nt^3\phi_{\max}^3\ln(1/n\varepsilon)^2}. \tag{18}$$

Now we can further derive

$$\phi(\hat{\pi}^t) - \phi(\hat{\pi}^0) \geq \frac{256}{6nt^3\phi_{\max}^3\ln(1/n\varepsilon)^2} - n\eta\phi_{\max}\varepsilon$$
$$\geq \frac{256}{8nt^3\phi_{\max}^3\ln(1/n\varepsilon)^2}. \tag{19}$$

The second inequality is hold by $\eta \leq \frac{1}{2n\phi_{\max}}$ and Eq. 18, which implies that

$$n\eta\phi_{\max}\varepsilon \leq \varepsilon/2 \leq \frac{256}{24nt^3\phi_{\max}^3\ln(1/n\varepsilon)^2}.$$

**Situation 2: (B) holds** Another situation is that we end the interval and start a communication round by update rule (B). In this case, we will have

$$t \leq \frac{1}{n^{1/4}\varepsilon^{1/4}\phi_{\max}\ln(1/n\varepsilon)} + 1 \leq \frac{2}{n^{1/4}\varepsilon^{1/4}\phi_{\max}\ln(1/n\varepsilon)},$$

which implies that

$$\varepsilon \leq \frac{16}{nt^4\phi_{\max}^4\ln(1/n\varepsilon)^4}. \tag{20}$$

Hence, we have

$$\phi(\hat{\pi}^t) - \phi(\hat{\pi}^0) \geq -n\eta\phi_{\max}t\varepsilon \geq -\frac{8}{nt^3\phi_{\max}^4\ln(1/n\varepsilon)^4} \geq -\frac{8}{nt^3\phi_{\max}^4\ln(1/n\varepsilon)^2}. \tag{21}$$

Since $\phi_{\max} \geq 1$, we can get

$$\frac{256}{8nt^3\phi_{\max}^3\ln(1/n\varepsilon)^2} \geq 2 \times \frac{8}{nt^3\phi_{\max}^4\ln(1/n\varepsilon)^2}.$$

**Communication Complexity** Now we turn to bound the number of communication rounds. Since $\phi(\hat{\pi}_h^T)$ is upper bounded by $\phi_{\max}$, suppose we run the algorithm for $T$ rounds and start new communication rounds at steps $0 = t_0 < t_1 < \cdots < t_{s^*} < T$. Let the number of communication rounds that terminate by rule (A) and rule (B) be denoted as $A^*$ and $B^*$ respectively.

First, if $A^* \geq B^*$ we have

$$\phi_{\max} \geq \phi(\hat{\pi}_h^{t_{s^*}}) - \phi(\hat{\pi}_h^0)$$

$$\geq \sum_{s=1}^{s^*} \frac{256}{8n(t_s - t_{s-1})^3\phi_{\max}^3\ln(1/n\varepsilon)^2} \cdot \mathbb{I}\left\{t_s - t_{s-1} \leq \frac{1}{n^{1/4}\varepsilon^{1/4}\phi_{\max}\ln(1/n\varepsilon)}\right\}$$

$$- \frac{8}{n(t_s - t_{s-1})^3\phi_{\max}^4\ln(1/n\varepsilon)^2} \cdot \mathbb{I}\left\{t_s - t_{s-1} \geq \frac{1}{n^{1/4}\varepsilon^{1/4}\phi_{\max}\ln(1/n\varepsilon)}\right\}$$

$$\geq \sum_{s=1}^{s^*} \frac{1}{2} \cdot \frac{256}{8n(t_s - t_{s-1})^3\phi_{\max}^3\ln(1/n\varepsilon)^2} \cdot \mathbb{I}\left\{t_s - t_{s-1} \leq \frac{1}{n^{1/4}\varepsilon^{1/4}\phi_{\max}\ln(1/n\varepsilon)}\right\}$$

$$+ \sum_{s=1}^{s^*} \frac{8}{n(t_s - t_{s-1})^3\phi_{\max}^4\ln(1/n\varepsilon)^2} \cdot \mathbb{I}\left\{t_s - t_{s-1} \leq \frac{1}{n^{1/4}\varepsilon^{1/4}\phi_{\max}\ln(1/n\varepsilon)}\right\}$$

$$- \sum_{s=1}^{s^*} \frac{8}{n(t_s - t_{s-1})^3\phi_{\max}^4\ln(1/n\varepsilon)^2} \cdot \mathbb{I}\left\{t_s - t_{s-1} \geq \frac{1}{n^{1/4}\varepsilon^{1/4}\phi_{\max}\ln(1/n\varepsilon)}\right\}$$

$$\geq \sum_{s=1}^{s^*} \frac{1}{2} \cdot \frac{256}{8n(t_s - t_{s-1})^3\phi_{\max}^4\ln(1/n\varepsilon)^2} \cdot \mathbb{I}\left\{t_s - t_{s-1} \leq \frac{1}{n^{1/4}\varepsilon^{1/4}\phi_{\max}\ln(1/n\varepsilon)}\right\}$$

$$+ \frac{8\varepsilon^{3/4}\ln(1/n\varepsilon)}{n^{1/4}\phi_{\max}} \cdot A^* - \frac{8\varepsilon^{3/4}\ln(1/n\varepsilon)}{n^{1/4}\phi_{\max}} \cdot B^*$$

$$\geq \frac{1}{2} \cdot \sum_{s=1}^{s^*} \frac{256}{8n(t_s - t_{s-1})^3\phi_{\max}^3\ln(1/n\varepsilon)^2} \cdot \mathbb{I}\left\{t_s - t_{s-1} \leq \frac{1}{n^{1/4}\varepsilon^{1/4}\phi_{\max}\ln(1/n\varepsilon)}\right\}$$

$$\geq \frac{256}{16n\phi_{\max}^3\ln(1/n\varepsilon)^2} \sum_{s=1}^{s^*} \frac{1}{(t_s - t_{s-1})^3} \cdot \mathbb{I}\left\{t_s - t_{s-1} \leq \frac{1}{n^{1/4}\varepsilon^{1/4}\phi_{\max}\ln(1/n\varepsilon)}\right\}. \quad (22)$$

Now denote $\mathbb{S}^* = \left\{s \in s^* \mid t_s - t_{s-1} \leq \frac{1}{n^{1/4}\varepsilon^{1/4}\phi_{\max}\ln(1/n\varepsilon)}\right\}$ as all the indexes of communication rounds that ends by rule (A). Hence, $|\mathbb{S}^*| \leq A^*$, and it is easy to show that

$$\sum_{s \in \mathbb{S}^*} (t_s - t_{s-1}) \leq T.$$

Now, by Cauchy's Inequality, we first know that

$$\sum_{s \in \mathbb{S}^*} \frac{1}{(t_s - t_{s-1})} \geq \frac{(A^*)^2}{\sum_{s \in \mathbb{S}^*}(t_s - t_{s-1})} \geq \frac{(A^*)^2}{T}.$$

Then, we can further get

$$T \cdot \left(\sum_{s \in \mathbb{S}^*} \frac{1}{(t_s - t_{s-1})^3}\right) \geq \left(\sum_{s \in \mathbb{S}^*} (t_s - t_{s-1})\right) \cdot \left(\sum_{s \in \mathbb{S}^*} \frac{1}{(t_s - t_{s-1})^3}\right)$$

$$\geq \left(\sum_{s \in \mathbb{S}^*} \frac{1}{t_s - t_{s-1}}\right)^2 \geq \frac{(A^*)^4}{T^2},$$

which implies that

$$\sum_{s=1}^{s^*} \frac{1}{(t_s - t_{s-1})^3} \cdot \mathbb{I}\left\{t_s - t_{s-1} \leq \frac{1}{n^{1/4}\varepsilon^{1/4}\phi_{\max}\ln(1/n\varepsilon)}\right\} = \sum_{s \in \mathbb{S}^*} \frac{1}{(t_s - t_{s-1})^3} \geq \frac{(A^*)^4}{T^3}. \quad (23)$$

Hence, by combining Eq. 22 and Eq. 23, we can finally get

$$\frac{16n\phi_{\max}^4\ln(1/n\varepsilon)^2}{256} \cdot T^3 \geq (A^*)^4,$$

which implies that the number of communication round is at most

$$s = A^* + B^* \leq 2A^* \leq \frac{2n^{1/4}\sqrt{\ln(1/n\varepsilon)} \cdot \phi_{\max}T^{3/4}}{4} = \widetilde{\mathcal{O}}\left(n^{1/4}\phi_{\max}T^{3/4}\right).$$

If $A^* \leq B^*$, we know that $s = A^* + B^* \leq 2B^*$. Moreover, it is obvious that

$$B^* \leq \frac{T}{\frac{1}{n^{1/4}\varepsilon^{1/4}\phi_{\max}\ln(1/n\varepsilon)}} = T \cdot n^{1/4}\varepsilon^{1/4}\phi_{\max}\ln(1/n\varepsilon).$$

Hence, we have

$$s \leq \max\left\{\widetilde{\mathcal{O}}\left(n^{1/4}\phi_{\max}T^{3/4}\right), T \cdot n^{1/4}\varepsilon^{1/4}\phi_{\max}\ln(1/n\varepsilon)\right\}. \quad (24)$$

By substituting $T = 2/\varepsilon$ into Eq. 24, we can get the communication round result.

**Convergence**   Now we consider the convergence guarantee. By Lemma A.2, we have

$$\phi(\hat{\pi}^{t+1}) - \phi(\hat{\pi}^t) \geq \phi(\pi^{t+1}) - \phi(\hat{\pi}^t) - n\eta\phi_{\max}\varepsilon$$

$$\geq (1 - \sqrt{n}\eta)\sum_{i\in[n]}\langle\pi_i^{t+1} - \hat{\pi}_i^t, \hat{\ell}_i(a, \hat{\pi}_{-i}^t)\rangle - n\eta\phi_{\max}\varepsilon$$

$$\geq (1 - \sqrt{n}\eta)\sum_{i\in[n]}\langle\pi_i^{t+1} - \hat{\pi}_i^t, \ell_i(a, \hat{\pi}_{-i}^t)\rangle - n\eta\phi_{\max}\varepsilon.$$

Now we introduce the Lemma 3.2 in (Sun et al., 2023), which helps us to complete the final proof.

**Lemma A.3.** *[Lemma 3.2 in (Sun et al., 2023)]*

$$
\max_{a_i \in \mathcal{A}_i} \langle a_i - \hat{\pi}_i^t, \ell_i(\cdot, \hat{\pi}_i^t) \rangle \geq \langle \pi_i^{t+1} - \hat{\pi}_i^t, \ell_i(\cdot, \hat{\pi}_i^t) \rangle \geq \max_{a_i \in \mathcal{A}_i} \langle a_i - \hat{\pi}_i^t, \ell_i(\cdot, \hat{\pi}_i^t) \rangle \cdot \left( 1 - \frac{1}{c\Delta\eta + 1} \right).
$$

*Proof.* The proof is exactly the same as Lemma 3.2 in (Sun et al., 2023). $\square$

Now by Lemma A.3, we have

$$
\begin{aligned}
\frac{1}{T-1} \sum_{t=1}^{T-1} \mathrm{Gap}(\hat{\pi}_i^t) &\leq \frac{1}{T-1} \sum_{t=1}^{T-1} \sum_{i=1}^{n} \frac{1}{1 - \frac{1}{c\Delta\eta+1}} \langle \pi_i^{t+1} - \hat{\pi}_i^t, \ell_i(a, \hat{\pi}_i^t) \rangle \\
&\leq \frac{1}{T-1} \sum_{t=1}^{T-1} \frac{1}{(1 - R_{\max}\sqrt{n}\eta)\frac{c\Delta\eta}{c\Delta\eta+1}} \left( \phi(\hat{\pi}^{t+1}) - \phi(\hat{\pi}^t) + n\eta\phi_{\max}\varepsilon \right) \\
&\leq \frac{2}{T-1} \sum_{t=1}^{T-1} \left( 1 + \frac{1}{c\Delta\eta} \right) \left( \phi(\hat{\pi}^{t+1}) - \phi(\hat{\pi}^t) + n\eta\phi_{\max}\varepsilon \right) \\
&\leq 4n\eta\phi_{\max}\varepsilon \cdot \left( 1 + \frac{2nM}{c\Delta} \right) + \frac{2}{T-1} \cdot \phi_{\max} \cdot \left( 1 + \frac{2nM}{c\Delta} \right) \\
&\leq \left( 2\varepsilon + \frac{2\phi_{\max}}{(T-1)} \right) \left( 1 + \frac{2nM}{c\Delta} \right) \\
&\leq 4\phi_{\max}\varepsilon \cdot \left( 1 + \frac{2nM}{c\Delta} \right) \\
&= 8n\varepsilon \cdot M^2 \left( 1 + \frac{1}{c\Delta} \right)
\end{aligned}
$$

Hence, the $\frac{1}{T} \sum_{t=1}^{T} \mathrm{NE}_{\mathrm{gap}}(\hat{\pi}_i^t)$ can be less than $8n\varepsilon \cdot \phi_{\max}(1 + \frac{1}{c\Delta})$ when $T = \frac{2}{\varepsilon} \geq \frac{1}{\varepsilon} + 1$. Hence, the iterative complexity is $T = \Theta(1/\varepsilon)$. Moreover, by substituting this into the inequality 24, we know that the number of communication round can be bounded by $\mathcal{O}(\phi_{\max}/\varepsilon^{3/4})$.

**Sample Complexity** Now we consider the sample complexity for this algorithm. In fact, for each communication round $t$ and each policy in de-duplicated set $\Pi_t$, we should collect $N = \mathcal{O}(M^2(\sum_{i=1}^n |\mathcal{A}_i|) \cdot \log(2/\delta)/\varepsilon^2)$ samples for each agent $i$ and $a_i \in \mathcal{A}_i$. Hence, it requires

$$
\mathcal{O}(M^2(\sum_{i=1}^{n} |\mathcal{A}_i|) \cdot \log(2/\delta)/\varepsilon^2) \cdot \max_t |\Pi_t|.
$$

Now we study the cardinality of the $\Pi_t$. Note that only communication round $t$ has this set. Without loss of generality, we assume that this communication round is 0. Hence,

$$
\tilde{\pi}_i^t(a) \propto \pi_i^0(a) \cdot \exp(\eta t \ell_i(a, \pi^0)),
$$

and $\Pi_0$ is the unique set of $\{\overline{\pi}^t\}_{t \in [T]}$. Recall that $\mathcal{A}_i^t = \{a \in \mathcal{A}_i \mid \tilde{\pi}_i^t(a) \leq \varepsilon/|\mathcal{A}_i|\}$ and

$$
\overline{\pi}_i^t(a_i) = \begin{cases} \frac{\varepsilon}{|\mathcal{A}_i|}, & \text{if } a \in \mathcal{A}_i^t \\ \left( 1 - \frac{\varepsilon|\mathcal{A}_i^t|}{|\mathcal{A}_i|} \right) \tilde{\pi}_i^t(a_i), & \text{else} \end{cases}. \tag{25}
$$

We now study the cardinality of $\Pi_0$. In fact, if there exists a $t_0$ such that $\tilde{\pi}_i^{t_0}(a) \leq \frac{\varepsilon}{|\mathcal{A}_i|}$ holds for all $i \in [n]$ and $a \notin \mathcal{A}_i^*$, we know that $\overline{\pi}_i^t$ will be the same for all $t \geq t_0$. Hence, $|\Pi_0| \leq t_0$ and we only need to find a upper bound for such a $t_0$.

Now we follow the Assumption 4.1. Define $a^* \in \mathcal{A}_i^* = \arg\max_{a \in \mathcal{A}_i} \ell_i(a, \pi^0)$ and $a^{**} = \arg\max_{a \in \mathcal{A}_i \setminus \{\mathcal{A}_i^*\}} \ell_i(a, \pi^0)$. Also, define $a_{\max} = \arg\max_{a \in \mathcal{A}_i^*} \pi^0(a)$. By Assumption 4.1, $\pi_i^0(a_{\max}) \geq \frac{1}{|\mathcal{A}_i^*|} \sum_{a \in \mathcal{A}_i^*} \pi_i^0(a) \geq \frac{c}{|\mathcal{A}_i|}$. Since the suboptimality gap $\Delta = \ell_i(a^*, \pi^0) -$

$\ell_i(a^{**}, \pi^0) > 0$, we know that $\frac{\tilde{\pi}_i^t(a)}{\pi_i^0(a)} \leq e^{-\eta\Delta}$ for $a \notin \mathcal{A}_i^*$. Hence, for $t \geq \frac{\log(|\mathcal{A}_i|^2/c\varepsilon)}{\eta\Delta}$, we have

$$\tilde{\pi}_i^t(a) \leq e^{-\eta\Delta} \cdot |\mathcal{A}_i|/c \leq \frac{\varepsilon}{|\mathcal{A}_i|}, \quad \forall a \notin \mathcal{A}_i^*,$$

which implies that

$$|\Pi_0| \leq \frac{\log(|\mathcal{A}_i|/\varepsilon)}{\eta\Delta} = \frac{2nM\log(|\mathcal{A}_i|/\varepsilon)}{\Delta}.$$

Hence, in general, we can get

$$\max_t |\Pi_t| \leq \frac{2nM\log(|\mathcal{A}_i|/\varepsilon)}{\Delta}.$$

and the sample complexity is no more than $\tilde{\Theta}\left(\frac{nM^3(\sum_{i=1}^n |\mathcal{A}_i|)\log(1/\delta)}{\varepsilon^{11/4}\Delta}\right)$, where $\tilde{\Theta}$ ignores the logarithm term $\log(|\mathcal{A}_i|/\varepsilon)$.

Finally, we prove that $g^t$ is a good estimate of the equilibrium gap of $\hat{\pi}^t$. First, by Eq. 8 and Eq. 17, we have

$$\sum_{i=1}^n \left(\mathbb{E}_{a_i \sim \hat{\pi}_i^{t+1}}[\hat{\ell}_i(a_i, \hat{\pi}_{-i}^t)] - \hat{\ell}_i(\hat{\pi}^t)\right) \geq \sum_{i=1}^n \left(\mathbb{E}_{a_i \sim \hat{\pi}_i^{t+1}}[\ell_i(a_i, \hat{\pi}_{-i}^t)] - \ell_i(\hat{\pi}^t) - 2\varepsilon\right) \quad (26)$$

$$\geq \sum_{i=1}^n \left(\mathbb{E}_{a_i \sim \pi_i^{t+1}}[\ell_i(a_i, \hat{\pi}_{-i}^t)] - \ell_i(\hat{\pi}^t) - 2\varepsilon - 4\eta n\varepsilon\right)$$

$$\geq \sum_{i=1}^n \langle \pi_i^{t+1} - \hat{\pi}_i^t, \ell_i(\cdot, \hat{\pi}^t)\rangle - 3n\varepsilon.$$

By Lemma A.3, we can further get

$$\text{Gap}(\hat{\pi}^t) \leq \left(1 + \frac{1}{c\Delta\eta}\right) \sum_{i=1}^n \langle \pi_i^{t+1} - \hat{\pi}_i^t, \ell_i(\cdot, \hat{\pi}^t)\rangle \leq \left(1 + \frac{1}{c\Delta\eta}\right)(g^t + 3n\varepsilon). \quad (27)$$

Similarly, we can derive $g^t \leq 3n\varepsilon + \sum_{i=1}^n \left(\mathbb{E}_{a_i \sim \hat{\pi}_i^{t+1}}[\hat{\ell}_i(a_i, \hat{\pi}_{-i}^t)] - \hat{\ell}_i(\hat{\pi}^t)\right) \leq 3n\varepsilon + \text{Gap}(\hat{\pi}^t)$. Hence, we finally get

$$g^t - 3n\varepsilon \leq \text{Gap}(\hat{\pi}^t) \leq \left(1 + \frac{1}{c\Delta\eta}\right)(g^t + 3n\varepsilon).$$

Now since

$$\min_{t \in [T-1]} \text{Gap}(\hat{\pi}^t) \leq \frac{1}{T-1} \sum_{t=1}^{T-1} \text{Gap}(\hat{\pi}^t) \leq 8n\varepsilon \cdot M^2 \left(1 + \frac{1}{c\Delta}\right),$$

denote $t^* = \arg\min_{t \in [T-1]} \text{Gap}(\hat{\pi}^t)$ and $k^* = \arg\min_{k \in [T-1]} g^k$, we know that

$$\text{Gap}(\hat{\pi}^{k^*}) \leq \left(1 + \frac{1}{c\Delta\eta}\right)(g^{k^*} + 3n\varepsilon)$$

$$\leq \left(1 + \frac{1}{c\Delta\eta}\right)(g^{t^*} + 3n\varepsilon)$$

$$\leq \left(1 + \frac{1}{c\Delta}\right)(\text{Gap}(\hat{\pi}^t) + 6n\varepsilon) \cdot \frac{1}{\eta}$$

$$\leq 28n\varepsilon \cdot M^3 \left(1 + \frac{1}{c\Delta}\right)^2 . \quad (28)$$

The last inequality uses the fact that $\eta = \frac{1}{2nM}$ Taking the union bound over all the concentration inequality, we know that the proof holds with probability at least $1 - p\delta$ with some constant $p$. Replace the failure probability $p\delta$ with $\delta$, we complete the proof.

$\square$

### A.1.1 PROOF OF COROLLARY 4.3

*Proof.* First recall that $s = \lfloor \log_2(T/2n) \rfloor$ and $I_s = (2n \cdot (2^{s-1} - 1), 2n \cdot (2^s - 1)]$. Hence, we have

$$\frac{4}{n\varepsilon_s} = 2^{s+1} \geq \frac{T}{2n} \geq 2^s = \frac{2}{n\varepsilon_s},$$

which implies that

$$\frac{4}{T} \leq \varepsilon_s \leq \frac{8}{T}$$

Since $\pi^{out}$ is the outputted policy of the oracle $PG(\mathcal{P}, 5n \times 2^{s-1}, \varepsilon_s)$, by Theorem 4.2, we know that

$$\text{Gap}(\pi^{\text{out}}) \leq 28n\varepsilon_s \cdot M^3 \left(1 + \frac{1}{c\Delta}\right)^2 \leq \frac{280nM^3(1 + \frac{1}{c\Delta})^2}{T}.$$

Now we consider the total sample complexity. The PG-oracle $PG(\mathcal{P}, n \times 2^i, \varepsilon_i)$ is called for at most $1 \leq i \leq s+1$. Hence, by the sample complexity result in Theorem 4.2, the total sample complexity is bounded by

$$\sum_{i=1}^{s+1} \widetilde{\mathcal{O}}\left(\frac{nM^3(\sum_{i=1}^n |\mathcal{A}_i|)\log(1/\delta)}{\varepsilon_i^{11/4}\Delta}\right) \leq \widetilde{\mathcal{O}}\left(\frac{nM^3(\sum_{i=1}^n |\mathcal{A}_i|)\log(1/\delta)}{\Delta}\right) \sum_{i=1}^{s+1} \frac{1}{\varepsilon_i^{11/4}}$$

$$\leq \widetilde{\mathcal{O}}\left(\frac{nM^3(\sum_{i=1}^n |\mathcal{A}_i|)\log(1/\delta)}{\Delta}\right) \frac{T^{11/4} \cdot (s+1)}{10^{11/4}}$$

$$\leq \widetilde{\mathcal{O}}\left(\frac{nM^3(\sum_{i=1}^n |\mathcal{A}_i|)\log(1/\delta)T^{11/4}}{\Delta}\right).$$

Similarly, the total communication cost is bounded by

$$\sum_{i=1}^{s+1} \widetilde{\mathcal{O}}\left(n^{1/4}\phi_{\max}\varepsilon_i^{-3/4}\right) = n^{1/4}\phi_{\max} \cdot \widetilde{\mathcal{O}}\left(\sum_{i=1}^{s+1} \varepsilon_i^{3/4}\right) = \widetilde{\mathcal{O}}(n^{1/4}\phi_{\max}T^{3/4}).$$

$\square$

## B   MCG ALGORITHMS

Now we introduce subprocedures in Algorithm 2 in detail.

**PG-Share**   The subprocedure PG-Share at the step $h$ takes the exploration policy $\overline{\pi}^t$, the value function $\{\overline{V}_{h+1}^{t+1}\}$ of the step $h+1$, and the number of rounds $K$ as the input. First, for each state-step pair $(s, h) \in \mathcal{S} \times [H]$, this subprocedure construct a potential game with true reward $r(s, \boldsymbol{a}) + \mathbb{E}_{s' \sim \mathbb{P}_h(s'|s,a)}\left[\overline{V}_{h+1}(s')\right]$. At each round $k$, all agents first execute the exploration policy $\overline{\pi}^t$ until step $h$ and observe the state $s_h^k$. Given this state, all agents use the PG-Unknown oracle for the potential game corresponding to the state $s_h$ to decide the action $\boldsymbol{a}_h^k$. Then, based on the state $s_{h+1}$, the datapoint $(\boldsymbol{a}_h^k, r(s_h^k, \boldsymbol{a}_h^k) + \overline{V}_{h+1}(s_{h+1}^k))$ are stored in the shared buffer, which served as an unbiased and bounded approximated reward for potential game $\mathcal{P}_{s_h}$. All agents update the policy and decide whether to start communication to get rewards for the potential game $\mathcal{P}_{s_h}$.

Intuitively, by constructing a potential game instance for each state $s \in \mathcal{S}$, we can compute the policy $\pi^{\text{out}}(\cdot \mid s)$ locally based on the corresponding potential game, where the number of executions of $\mathcal{P}_s$ is fully determined by how frequently the exploration policy visits state $s_h$ at step $h$. Leveraging Corollary 4.3, we can obtain per-state error bounds for the policy $\pi^{\text{out}}$, which enables us to use optimistic V-learning to promote exploration of underexplored states. The following theorem shows the final theoretical guarantee of our PG-share subprocedure.

**Lemma B.1.** *Denote the output policy of the Algorithm 3 PG-Share$_h(\overline{\pi}, \overline{V}_{h+1}, \sqrt{t})$ with step $h$ as $\pi_h^{out}$. Then, with probability at least $1 - \delta$, for any state $s \in \mathcal{S}$ we have*

$$\max_{\mu_{i,h} \in \Delta(\mathcal{A}_i)} (\mathbb{E}_{\mu_{i,h} \times \pi_{-i,h}^{out}} - \mathbb{E}_{\pi_h^{out}}) \left[r_h + \mathbb{P}_{h+1}\overline{V}_{h+1}\right](s) \leq \overline{G}(\overline{\pi}^t, t),$$

---

**Algorithm 3** PG-Share$_h(\overline{\pi}, \overline{V}_{h+1}, K)$

---

1: Construct Potential Game $\mathcal{P}_{s,h}$ with reward $r(s, \boldsymbol{a}) + \mathbb{E}_{s' \sim \mathbb{P}_h(s'|s,a)} \left[ \overline{V}_{h+1}(s') \right]$ for each state $s \in \mathcal{S}$.
2: **for** $k = 1, 2, \cdots, K$ **do**
3:    Execute the policy $\overline{\pi}_{1:h-1}$ to get state $s_h^k$.
4:    Play the $s_h^k$-th game PG$_{s_h}$ once: Take action $\boldsymbol{a}_h^k \sim \pi_h^k(\cdot \mid s_h^k)$ and get $s_{h+1}$.
5:    The buffer receive the datapoint $(\boldsymbol{a}_h^k, r(s_h^k, \boldsymbol{a}_h^k) + \overline{V}_{h+1}(s_{h+1}^k))$. Update the policy $\pi_h^{k+1}(\cdot \mid s)$ and decide whether to start a communication round based on PG-Unknown($\mathcal{P}_{s_h}$).
6: **end for**
7: **Return** $\pi^{\text{out}}(\cdot \mid s) = \text{PG-Unknown}(\mathcal{P}_s)$.

---

**Algorithm 4** V-Approx$_h(\overline{\pi}, \pi_h, \overline{V}_{h+1}, t)$

---

1: Execute the policy $\overline{\pi}_{1:h-1} \circ (\pi_h)$ for $t$ times to collect $\{s_h^q, \boldsymbol{a}_h^q, \hat{r}_h^q, s_{h+1}^q\}_{q \in [K]}$.
2: Start a communication round to get all rewards and trajectories.
3: Calculate $\overline{V}_h(s) = \frac{1}{N_h(s)} \sum_{q \in [K]} [\hat{r}_h^q(s_h^q, \boldsymbol{a}_h^q) + \overline{V}_{h+1}(s_{h+1})] + 3G(N_h(s), t)$, where $G(N_h(s), t) = \frac{280nH^3(1+\frac{1}{c\Delta})^2}{N_h(s)/\sqrt{K} + \tau/2}$.
4: **Return** $\overline{V}_h(s)$.

---

where $\overline{G}(\overline{\pi}^t, t) = \frac{280nH^3(1+\frac{1}{c\Delta})^2 \cdot \tau}{\sqrt{t} \cdot \mathbb{P}_{\overline{\pi}^t}(s_h = s) + \sqrt{\tau/2}}$ and $\tau = \log(S \max_{i \in [n]} |\mathcal{A}_i| t/\delta) \geq 2$ *only contains logarithm terms.*

**V-Approx**    Now we analyze the V-Approx subprocedure. The V-Approx subprocedure takes the exploration policy $\overline{\pi}$, the current policy $\pi_h$, the value function at the step $h + 1$, and the number of samples $K$ as input. First, all agents execute the exploration policy $\overline{\pi}^t$ for step $1, 2, \cdots, h - 1$ and policy $\pi_h$ for step $h$. Then, all agents start a communication round to collect all trajectories. Finally, each agent calculates the value function of step $h$ by Line 3 in Algorithm 4. Line 3 constructs an optimistic estimate of the value function by adding the bonus term $G(N_h(s), t)$, in order to encourage exploration for the underexplored states. As the following theorem shows, this bonus term gives us an optimistic estimate, which is required for the final theoretical proof.

**Lemma B.2** (V-Approx-Share). *From Algorithm 3 V-Approx$_h(\overline{\pi}, \pi_h, \overline{V}_{h+1}, t)$, with probability at least $1 - \delta$, we have*

$$\overline{V}_h(s) \geq \mathbb{E}_{\pi_h} \left[ r_h(s, a) + \mathbb{P}_{h+1} \overline{V}_{h+1} \right](s) + \overline{G}(\overline{\pi}^t, t).$$
$$\overline{V}_h(s) \leq \mathbb{E}_{\pi_h} \left[ r_h(s, a) + \mathbb{P}_{h+1} \overline{V}_{h+1} \right](s) + 7\overline{G}(\overline{\pi}^t, t),$$

**Trigger Condition**    One critical component of Algorithm 2 is the trigger condition, which ensures that PG-Share and V-Approx are not called at every round $t$, reducing unnecessary computation and communication. We implement the trigger condition using the doubling trick Wang et al. (2023). Specifically, the condition is satisfied when there exists a step $h$ and a state $s \in \mathcal{S}$ such that $T_h^t(s) \geq 2T_h^{I_t}(s)$, where $I_t$ denotes the most recent round in which the policy was updated. This helps us to reduce the sample complexity and the number of communication rounds without increasing the estimation error.

## B.1   PROOF OF LEMMA B.1

*Proof.* We analysis the algorithm PG-Share$_h(\overline{\pi}, \overline{V}_{h+1}, \lfloor \sqrt{t} \rfloor + 1)$. First, denote $N_h(s)$ as the number of times that agents visit $s$ for $\sqrt{t}$ trajectories, i.e.,

$$N_h(s) = \sum_{k=1}^{\lfloor \sqrt{t} \rfloor + 1} \mathbb{I}\{s_h^k = s\},$$

where $s_h^k$ is defined in Line 3 in Algorithm 3.

Now by Corollary 4.3 and taking a union bound over all potential game instances constructed in Algorithm 3, for any state $s \in \mathcal{S}$, we can have

$$\max_{\mu_{i,h} \in \Delta(\mathcal{A}_i)} (\mathbb{E}_{\mu_{i,h} \times \pi_{-i,h}} - \mathbb{E}_{\pi_h}) \left[ r(s, \boldsymbol{a}) + \mathbb{E}_{s' \sim \mathbb{P}_h(s'|s,a)}[\overline{V}_{h+1}(s')] \right] \leq \frac{280 n H^3 (1 + \frac{1}{c\Delta})^2}{N_h(s)}, \quad (29)$$

where the inequality holds because the reward for the potential game $\mathcal{P}_{s,h}$ is $r(s, \boldsymbol{a}) + \mathbb{E}_{s' \sim \mathbb{P}_h(s'|s,a)}[\overline{V}_{h+1}(s')] \leq H$ and the maximum value of the potential function is $\phi_{\max} \leq H$. Now, by Bernstein inequality and taking a union bound for all $s \in \mathcal{S}$, with probability at least $1 - \delta$, we have

$$\frac{1}{2}\sqrt{t} \cdot \mathbb{P}_{\overline{\pi}}(s_h = s) - \frac{1}{2}\tau \leq N_h(s) \leq 2\sqrt{t} \cdot \mathbb{P}_{\overline{\pi}}(s_h = s) + \frac{1}{2}\tau, \quad (30)$$

where $\tau = \log(S \max_{i \in [n]} |\mathcal{A}_i| T/\delta) \geq 2$. Hence, combining Eq. 29, we have

$$\max_{\mu_{i,h} \in \Delta(\mathcal{A}_i)} (\mathbb{E}_{\mu_{i,h} \times \pi_{-i,h}} - \mathbb{E}_{\pi_h}) \left[ r(s, \boldsymbol{a}) + \mathbb{E}_{s' \sim \mathbb{P}_h(s'|s,a)}[\overline{V}_{h+1}(s')] \right] \quad (31)$$

$$\leq \frac{280 n H^3 (1 + \frac{1}{c\Delta})^2}{\max\{1, \sqrt{t} \cdot \mathbb{P}_{\overline{\pi}^t}(s_h = s) - \tau/2\}} \quad (32)$$

$$\leq \frac{280 n H^3 (1 + \frac{1}{c\Delta})^2 \cdot \tau}{\sqrt{t} \cdot \mathbb{P}_{\overline{\pi}^t}(s_h = s) + \tau/2} \quad (33)$$

$$\leq \frac{280 n H^3 (1 + \frac{1}{c\Delta})^2 \cdot \tau}{\sqrt{t} \cdot \mathbb{P}_{\overline{\pi}^t}(s_h = s) + \sqrt{\tau/2}} \quad (34)$$

$$= \overline{G}(\overline{\pi}^t, t). \quad (35)$$

The last inequality holds for $\tau \geq 2$ with some algebra. $\qquad \square \qquad\qquad \square$

## B.2 Proof of Lemma B.2

*Proof.* By Hoeffding's inequality, since $\overline{V}_h(s)$ is an unbiased estimator with upper bound $H$, we can know that

$$\left| \overline{V}_h(s) - 3G(N_h(s), t) - \mathbb{E}_{\pi_h}[r_h(s, a) + \mathbb{P}_{h+1}\overline{V}_{h+1}](s) \right| \leq \frac{H\sqrt{\log(2/\delta)}}{\sqrt{N_h(s)}}.$$

By Eq. 30, we have

$$\left| \overline{V}_h(s) - 3G(N_h(s), t) - \mathbb{E}_{\pi_h}[r_h(s, a) + \mathbb{P}_{h+1}\overline{V}_{h+1}](s) \right|$$

$$\leq \frac{H\sqrt{\log(2/\delta)}}{\sqrt{\max\{1, t \cdot \mathbb{P}_{\overline{\pi}^t}(s_h = s) - \frac{1}{2}\tau\}}}$$

$$\leq \frac{n H^3 (1 + \frac{1}{c\Delta})^2 \cdot \tau}{\sqrt{t \cdot \mathbb{P}_{\overline{\pi}^t}(s_h = s) + \frac{1}{2}\tau}}.$$

Also, we have

$$G(N_h(s), t) = \frac{280 n H^3 (1 + \frac{1}{c\Delta})^2 \tau}{N_h(s)/\sqrt{t} + \tau/2} \leq \frac{280 n H^3 (1 + \frac{1}{c\Delta})^2 \tau}{\frac{1}{2}\sqrt{t} \cdot \mathbb{P}_{\overline{\pi}_t}(s_h = s) + \tau/4}. \quad (36)$$

The last inequality holds since $t \geq 4$. Similarly, we have

$$G(N_h(s), t) \geq \frac{280 n H^3 (1 + \frac{1}{c\Delta})^2 \tau}{2\sqrt{t} \cdot \mathbb{P}_{\overline{\pi}_t}(s_h = s) + \tau}. \quad (37)$$

Now we turn to prove our lemma. By Eq. 36, we have

$$\overline{V}_h(s) \leq \mathbb{E}_{\pi_h}[r_h(s, a) + \mathbb{P}_{h+1}\overline{V}_{h+1}](s) + \frac{n H^3 (1 + \frac{1}{c\Delta})^2 \cdot \tau}{\sqrt{t \cdot \mathbb{P}_{\overline{\pi}^t}(s_h = s) + \frac{1}{2}\tau}} + 3G(N_h(s), t)$$

$$\leq \mathbb{E}_{\pi_h}[r_h(s,a) + \mathbb{P}_{h+1}\overline{V}_{h+1}](s) + \frac{2nH^3(1+\frac{1}{c\Delta})^2 \cdot \tau}{\sqrt{t \cdot \mathbb{P}_{\overline{\pi}^t}(s_h = s)} + \sqrt{\tau/2}} + \frac{840nH^3(1+\frac{1}{c\Delta})^2\tau}{\frac{1}{2}\sqrt{t} \cdot \mathbb{P}_{\overline{\pi}^t}(s_h = s) + \tau/4}$$

$$\leq \mathbb{E}_{\pi_h}[r_h(s,a) + \mathbb{P}_{h+1}\overline{V}_{h+1}](s) + \frac{1682nH^3(1+\frac{1}{c\Delta})^2\tau}{\sqrt{t} \cdot \mathbb{P}_{\overline{\pi}^t}(s_h = s) + \sqrt{\tau/2}}$$

$$\leq \mathbb{E}_{\pi_h}[r_h(s,a) + \mathbb{P}_{h+1}\overline{V}_{h+1}](s) + 7\overline{G}(\overline{\pi}^t, t).$$

Similarly, by Eq. 37, we have

$$\overline{V}_h(s) \geq \mathbb{E}_{\pi_h}[r_h(s,a) + \mathbb{P}_{h+1}\overline{V}_{h+1}](s) + \frac{840nH^3(1+\frac{1}{c\Delta})^2\tau}{2\sqrt{t} \cdot \mathbb{P}_{\overline{\pi}^t}(s_h = s) + \tau} - \frac{nH^3(1+\frac{1}{c\Delta})^2\tau}{\sqrt{t \cdot \mathbb{P}_{\overline{\pi}^t}(s_h = s) + \frac{1}{2}\tau}}$$

$$\geq \mathbb{E}_{\pi_h}[r_h(s,a) + \mathbb{P}_{h+1}\overline{V}_{h+1}](s) + \frac{420nH^3(1+\frac{1}{c\Delta})^2\tau}{\sqrt{t} \cdot \mathbb{P}_{\overline{\pi}^t}(s_h = s) + \tau/2} - \frac{2nH^3(1+\frac{1}{c\Delta})^2\tau}{\sqrt{t \cdot \mathbb{P}_{\overline{\pi}^t}(s_h = s)} + \sqrt{\tau/2}}$$

$$\geq \mathbb{E}_{\pi_h}[r_h(s,a) + \mathbb{P}_{h+1}\overline{V}_{h+1}](s) + \frac{418nH^3(1+\frac{1}{c\Delta})^2\tau}{\sqrt{t} \cdot \mathbb{P}_{\overline{\pi}^t}(s_h = s) + \sqrt{\tau/2}}$$

$$\geq \mathbb{E}_{\pi_h}[r_h(s,a) + \mathbb{P}_{h+1}\overline{V}_{h+1}](s) + \overline{G}(\overline{\pi}^t, t).$$

Hence we complete the proof. $\qquad\qquad\qquad\qquad\square$ $\qquad\qquad\qquad\qquad\square$

### B.3  MAIN FRAMEWORK

This strategy of choosing the exploration policy can guarantee sufficient coverage and upper bound the error of the estimation in expectation. The following theorem shows this statement in detail.

**Lemma B.3.** *For any policy sequence $\pi_1, \pi_2, \cdots, \pi_T$, we have*

$$\sum_{t=0}^{T-1}\sum_{h=1}^{H}\mathbb{E}_{s_h \sim \pi^{t+1}}\left[\overline{G}(\overline{\pi}^t, t)\right] \leq \widetilde{\mathcal{O}}\left(nSH^4\left(1 + \frac{1}{c\Delta}\right)^2\sqrt{T}\right).$$

*Proof.* Note that

$$\sum_{t=0}^{T-1}\sum_{h=1}^{H}\mathbb{E}_{s_h \sim \pi^t}\left[\overline{G}(\overline{\pi}^t, t)\right]$$

$$\leq 68nH^3(1 + \frac{1}{c\Delta})^2\tau \cdot \sum_{t=0}^{T-1}\sum_{h=1}^{H}\sum_{s \in \mathcal{S}}\frac{\sqrt{t} \cdot \mathbb{P}_{\pi^{t+1}}(s_h = s)}{t \cdot \mathbb{P}_{\overline{\pi}^t}(s_h = s) + \tau/2}$$

$$= 68nH^3(1 + \frac{1}{c\Delta})^2\tau \cdot \sum_{t=0}^{T-1}\sum_{h=1}^{H}\sum_{s \in \mathcal{S}}\frac{\sqrt{t} \cdot \mathbb{P}_{\pi^{t+1}}(s_h = s)}{\sum_{t'=1}^{t} \cdot \mathbb{P}_{\overline{\pi}^{t'}}(s_h = s) + 1}$$

Now we provide a lemma that can bound both terms (A) and (B).

**Lemma B.4.** *For any non-negative sequence $a_1, \cdots, a_T \in [0, 1]$, we have*

$$\sum_{t=0}^{T-1}\frac{a_{t+1}}{\sum_{i=1}^{t}a_i + 1} \leq 4\log T.$$

*Proof.* In fact, if we set $b_t = \frac{a_t}{\sum_{i=1}^{t-1}\mathcal{A}_i + 1} \leq 1$, we know $e^{b_t}/4 \leq 1 + b_t$ and $(1/4)^T \prod_{t=1}^{T}e^{b_t} \leq \prod_{t=1}^{T}(1 + b_t) \leq (a_1 + a_2 + \cdots + a_T) \leq T$, which implies $\sum_{t=1}^{T}b_t \leq 4\log T.$ $\qquad\square$

Hence, by Lemma B.4, we have

$$\sum_{t=1}^{T}\sum_{h=1}^{H}\mathbb{E}_{s_h \sim \pi^t}\left[\overline{G}(\overline{\pi}^t, t)\right] \leq 68nH^3(1 + \frac{1}{c\Delta})^2\tau\sqrt{T} \cdot SH \cdot 4\log T$$

$$= \widetilde{\mathcal{O}} \left( nSH^4 \left( 1 + \frac{1}{c\Delta} \right)^2 \sqrt{T} \right),$$

where $\widetilde{\mathcal{O}}$ ignores all logarithm terms in $S, H, \max_i |\mathcal{A}_i|, 1/\delta, T$. We complete the proof. $\qquad\square$

**Final Proof**  First, we assume $\pi^{t+1}$ updates at each timestep, without considering the trigger condition. By the definition of V-function and the definition of the best response, we know that

$$\max_{\mu^*_{i,h} \in \Delta(\mathcal{A}_i)} \mathbb{E}_{\mu^*_{i,h} \times \pi_{-i,h}} \left[ r_h + \mathbb{P}_{h+1} V^{\dagger, \pi_{-i}}_{h+1} \right] (s) = V^{\dagger, \pi_{-i}}_h(s).$$

Hence, by the theoretical guarantee of PG-share Theorem B.1, for any $t \geq 0$ we can easily know that

$$\max_{\mu_{i,h} \in \Delta(\mathcal{A}_i)} \mathbb{E}_{\mu_{i,h} \times \pi^{t+1}_{-i,h}} \left[ r_h + \mathbb{P}_{h+1} \overline{V}^{t+1}_{h+1} \right] (s)$$

$$\leq \mathbb{E}_{\pi^{t+1}_h} \left[ r_h + \mathbb{P}_{h+1} \overline{V}^{t+1}_{h+1} \right] (s) + \overline{G}(\overline{\pi}^t, t)$$

$$\leq \overline{V}^{t+1}_h(s).$$

The last inequality holds from Lemma B.2. Hence, by a simple recursion we have

$$\overline{V}^{t+1}_h(s) \geq V^{\dagger, \pi_{-i}}_h(s), \ \forall h \in [H].$$

Similarly, since

$$\mathbb{E}_{\pi^{t+1}_h} \left[ r_h + \mathbb{P}_{h+1} V^{\pi^{t+1}_{h+1}}_{h+1} \right] (s) = V^{\pi^{t+1}}_h(s),$$

by Lemma B.2 and a simple recursion, we can have

$$\overline{V}^{t+1}_h(s) \leq V^{\pi^{t+1}}_h(s) + 7 \sum_{h'=h}^{H} \mathbb{E}_{\pi^{t+1}} \left[ \overline{G}(\overline{\pi}^t, t) \right],$$

which implies that

$$\sum_{i=1}^{n} \sum_{t=1}^{T} V^{\dagger, \pi^t_{-i}}_1(s_1) - V^{\pi^t}_1(s_1) \leq 7 \sum_{i=1}^{n} \sum_{t=1}^{T} \sum_{h=1}^{H} \mathbb{E}_{\pi^t} \left[ \overline{G}(\overline{\pi}^{t-1}, t)) \right] \tag{38}$$

$$\leq \widetilde{\mathcal{O}} \left( n^2 SH^4 \cdot \left( 1 + \frac{1}{c\Delta} \right)^2 \sqrt{T} \right).$$

Now we consider the trigger condition. In fact, by Bernstein' inequality, with probability at least $1 - \delta$, we know that

$$\frac{1}{2} t \cdot \mathbb{P}_{\overline{\pi}^t}(s_h = s) - \frac{1}{2}\tau \leq T^t_h(s) \leq 2t \cdot \mathbb{P}_{\overline{\pi}^t}(s_h = s) + \frac{1}{2}\tau.$$

Now suppose $t' = \min\{t' \geq I_t \mid T^{t'}_h(s) \geq T^{I_t}_h(s)\}$, which is the minimum time step that satisfies the trigger condition. Hence, we have

$$\sqrt{t'} \mathbb{P}_{\overline{\pi}^{t'}}(s_h = s) + \sqrt{\tau/2} \leq \frac{2 T^{t'}_h(s) + 2\tau}{\sqrt{t}} + \sqrt{\tau/2}$$

$$\leq \frac{4 T^{I_t}_h(s)}{\sqrt{t}} + 2\tau + 2\sqrt{\tau/2}$$

$$\leq 8 \left( \sqrt{I_t} \cdot \mathbb{P}_{\overline{\pi}^{I_t}}(s_h = s) + \sqrt{\tau/2} \right).$$

Now since we check the trigger condition when $t - I_t = 2^a$ by the doubling trick, we know that $t - I_t \leq 2(t' - I_t)$. Hence,

$$t\mathbb{P}_{\overline{\pi}^t}(s_h = s) = \sum_{j=1}^{t} \mathbb{P}_{\pi^j}(s_h = s) \tag{39}$$

$$= \sum_{j=1}^{I_t} \mathbb{P}_{\pi^j}(s_h = s) + (t - I_t)\mathbb{P}_{\pi^{I_t+1}}(s_h = s). \tag{40}$$

The second equation is because $\pi^j = \pi^{I_t}$ for all $j \in [I_t + 1, t]$ since it is not been updated. Now we can further bound it by

$$t\mathbb{P}_{\overline{\pi}^t}(s_h = s) = \sum_{j=1}^{I_t} \mathbb{P}_{\pi^j}(s_h = s) + 2(t' - I_t)\mathbb{P}_{\pi^{I_t+1}}(s_h = s) \tag{41}$$

$$\leq 2 \times \left( \sum_{j=1}^{I_t} \mathbb{P}_{\pi^j}(s_h = s) + (t' - I_t)\mathbb{P}_{\pi^{I_t+1}}(s_h = s) \right) \tag{42}$$

$$= 2t'\mathbb{P}_{\overline{\pi}^{t'}}(s_h = s). \tag{43}$$

Hence, we have

$$\sqrt{t}\mathbb{P}_{\overline{\pi}^t}(s_h = s) + \sqrt{\tau/2} \leq \sqrt{2t'} \cdot 2\mathbb{P}_{\overline{\pi}^{t'}}(s_h = s) + \sqrt{\tau/2}$$

$$\leq 2\sqrt{2}\left( \sqrt{t'}\mathbb{P}_{\overline{\pi}^{t'}}(s_h = s) + \sqrt{\tau/2} \right)$$

$$\leq 16\sqrt{2}\left( \sqrt{I_t} \cdot \mathbb{P}_{\overline{\pi}^{I_t}}(s_h = s) + \sqrt{\tau/2} \right),$$

which implies that

$$\overline{G}(\overline{\pi}^{I_t}, I_t) \leq 16\sqrt{2} \cdot \overline{G}(\overline{\pi}^t, t).$$

Now follow the same proof process to by Eq. 38, we can get

$$\sum_{i=1}^n \sum_{t=1}^T V_1^{\dagger, \pi_{-i}^t}(s_1) - V_1^{\pi^t}(s_1) = \sum_{i=1}^n \sum_{t=1}^T V_1^{\dagger, \pi_{-i}^{I_t+1}}(s_1) - V_1^{\pi^{I_t+1}}(s_1)$$

$$\leq 7 \sum_{i=1}^n \sum_{t=1}^T \sum_{h=1}^H \mathbb{E}_{\pi^{I_t+1}}[\overline{G}(\overline{\pi}^{I_t+1}, I_t + 1)]$$

$$\leq 102\sqrt{2} \cdot \sum_{i=1}^n \sum_{t=1}^T \sum_{h=1}^H \mathbb{E}_{\pi^{I_t+1}}[\overline{G}(\overline{\pi}^t, t)]$$

$$= 102\sqrt{2} \cdot \sum_{i=1}^n \sum_{t=1}^T \sum_{h=1}^H \mathbb{E}_{\pi^t}[\overline{G}(\overline{\pi}^t, t)]$$

$$= \widetilde{\mathcal{O}}\left( n^2 S H^4 \cdot \left( 1 + \frac{1}{c\Delta} \right)^2 \sqrt{T} \right).$$

Letting $T = 1/\varepsilon^2$, we know that

$$\frac{1}{T} \sum_{t=1}^T \text{Gap}(\pi^t) \leq \widetilde{\mathcal{O}}\left( n^2 S H^4 \cdot \left( 1 + \frac{1}{c\Delta} \right)^2 \right) \cdot \varepsilon.$$

The final step of Algorithm 2 is to estimate the equilibrium gap and find the policy with a minimum gap. In fact, by collect $\mathcal{O}(H^2 \log(1/\delta)/\varepsilon^2)$ trajectories to estimate one value function, we can bound the estimation error by Hoeffding's inequality as

$$\left| \hat{V}^\pi(s_1) - V^\pi(s_1) \right| \leq \varepsilon,$$

$$\left| \hat{V}^{a_i \times \pi_{-i}}(s_1) - V^{a_i \times \pi_{-i}}(s_1) \right| \leq \varepsilon, \ \forall a_i \in \mathcal{A}_i,$$

which implies that

$$\text{Gap}(\pi^t) \leq \sum_{i=1}^n \max_{a_i \in \mathcal{A}_i} \left( V^{a_i \times \pi_{-i}^t}(s_1) - V^{\pi^t}(s_1) \right)$$

$$\leq \sum_{i=1}^{n} \max_{a_i \in \mathcal{A}_i} \left( \hat{V}^{a_i \times \pi_{-i}^t}(s_1) - \hat{V}^{\pi^t}(s_1) \right) + 2n\varepsilon$$

$$\leq g^t + 2n\varepsilon$$

Hence, we have

$$\text{Gap}(\pi^{t^*}) \leq g(t^*) + 2n\varepsilon \leq \frac{1}{T}\sum_{t=1}^{T} g(t) + 2n\varepsilon \tag{44}$$

$$\leq \frac{1}{T}\sum_{t=1}^{T} \text{Gap}(\pi^t) + 2n\varepsilon \tag{45}$$

$$\leq \widetilde{\mathcal{O}}\left( n^2 S H^4 \cdot \left(1 + \frac{1}{c\Delta}\right)^2 \right) \cdot \varepsilon. \tag{46}$$

We complete the proof.

**Communication Rounds** Now we calculate the number of communication rounds. For checking the trigger condition, it contains $\log(T) = \log(1/\varepsilon^2)$ communication rounds. Note that trigger condition can only hold for no more than $SH \log(T) = SH \log(1/\varepsilon^2)$ rounds.

At each time that trigger condition is satisifed, by Corollary 4.3, PG-Share Algorithm will take $\mathcal{O}\left( SH \cdot H(\sqrt{T})^{3/4} \right) = \mathcal{O}\left( SH^2 T^{3/8} \right) = \mathcal{O}\left( \frac{SH^2}{\varepsilon^{3/4}} \right)$ communication rounds since all PG-Share Oracles contain $SH$ potential game instances.

Also, V-Approx algorithm will take 1 communication rounds. Hence, these two algorithms need $\mathcal{O}\left( \frac{SH^2 \log(1/\varepsilon^2)}{\varepsilon^{3/4}} \right)$ rounds in total. Moreover, estimating the equilibrium gap needs at most $\mathcal{O}(SH \log T)$ communication rounds. This is because the number of distinct policies in $\{\pi^t\}_{t\in[T]}$ is at most $\mathcal{O}(SH \log T)$ by our trigger condition. Hence, it totally needs $\widetilde{\mathcal{O}}\left( \frac{S^2 H^3}{\varepsilon^{3/4}} \right)$ communication rounds.

**Sample Complexity** At each time that trigger condition is satisifed, by Corollary 4.3, all PG-Share oracles takes no more than $\widetilde{\mathcal{O}}\left( \frac{SH \cdot H^3 (\sum_{i=1}^{n} |\mathcal{A}_i|)\sqrt{T}^{11/4}}{\Delta} \right)$ since all oracles contain $SH$ potential game instances. Since the trigger condition happens for at most $\widetilde{\mathcal{O}}(SH)$ times, the total sample complexity for PG-Share oracles will be no more than $\widetilde{\mathcal{O}}\left( \frac{S^2 H^5 \cdot (\sum_{i=1}^{n} |\mathcal{A}_i|)\sqrt{T}^{11/4}}{\Delta} \right) = \widetilde{\mathcal{O}}\left( \frac{S^2 H^5 \cdot (\sum_{i=1}^{n} |\mathcal{A}_i|)}{\varepsilon^{11/4}\Delta} \right)$. For algorithm V-Approx, it takes at most $T \cdot H$ samples for each time. Hence, the number of total samples is at most $\widetilde{\mathcal{O}}(H^2 T) = \widetilde{\mathcal{O}}\left( \frac{H^2}{\varepsilon^2} \right)$. Finally, to estimate the equilibrium gap, we need $\widetilde{\mathcal{O}}\left( \frac{H^2 (\sum_{i=1}^{n} |\mathcal{A}_i|) \cdot \log(1/\delta)}{\varepsilon^2} \cdot SH \log(1/\varepsilon^2) \right) = \widetilde{\mathcal{O}}\left( \frac{SH^3 (\sum_{i=1}^{n} |\mathcal{A}_i|) \cdot \log(1/\delta)}{\varepsilon^2} \right)$ samples.

Hence, in total, the sample complexity will be no more than $\widetilde{\mathcal{O}}\left( \frac{S^2 H^5 \cdot (\sum_{i=1}^{n} |\mathcal{A}_i|)}{\varepsilon^{11/4}\Delta} \right)$. $\qquad\square$

## C  ENVIRONMENT DETAILS

### C.1  POTENTIAL GAMES

Our environment for potential game is a fully cooperative environment, in which all agents share a same reward $r(\boldsymbol{a})$. The number of agents is 3, while the number of actions for each agent is 10. We assign rewards for each action randomly sampled from the interval $[0, 0.2]$. The learning rate of all algorithms are set to 0.1.

We evaluate all algorithms over $N = 5000$ episodes, where agents access the reward buffer once every 500 episodes. For the base policy prediction method, we precompute five base policies

$\pi_1, \cdots, \pi_5$ by performing old gradient descent in the communication round and use $\pi_i$ for the base policy of the importance sampling for the next $[100 * (i - 1), 100i - 1]$ episodes. In all three algorithms, 100 samples are collected per communication round. In particular, in the base policy prediction algorithm, this corresponds to 20 samples per base policy. The standard NPG uses 20 episodes for each round and directly gets access to the reward at each timestep, inducing 5000 communication rounds. All results are averaged over five random seeds, with error bars indicating the standard deviation.

## C.2 CONGESTION GAMES

In this experiment, we evaluate our algorithms in a congestion game, which borrows the same experiment setting as (Sun et al., 2023). To be more specific, we consider $n = 8$ agents, each with an action space $A_i = A, B, C, D$. The environment consists of two states, $S = $ safe, distancing. An agent's reward for selecting action $a$ is given by $w_a^s$ times the number of other agents choosing the same action, where the weights satisfy $w_A^s < w_B^s < w_C^s < w_D^s$. Rewards in the distancing state are uniformly reduced relative to the safe state. The state transition depends on joint actions: if more than half of the agents select the same action, the system transitions to the distancing state; if actions are evenly distributed, it transitions to the safe state.

We run 500 episodes in total, and agents access the reward buffer every 30 episodes. For the base policy prediction method, we predict six base policies and use each for 5 episodes. All results are averaged over four random seeds, with error bars indicating the standard deviation.

## C.3 MAPPO

In this experiment, the environments are MPE (Lowe et al., 2017) and SMAC environments. For MPE environments, we refer (Lowe et al., 2017; Yu et al., 2022) for the detailed introduction. For SMAC environments, we refer (Samvelyan et al., 2019) for the detailed introduction. For both experiments, we run 5 random seeds, with error bars indicating the standard deviation.

At each timestep, we perform 10 gradient updates, following the standard MAPPO setup. We define a communication interval parameter $I$. In our BPP algorithm, at each communication round, we precompute $I$ policies $\pi_1, \cdots, \pi_I$ for the next $I$ steps, by performing the gradient descent using the gradient in the communication round. Then, for the next $I$ steps, we use these $I$ base policies as the base policy of the importance sampling, to get the new gradient.

**MPE** In the MPE experiment, for the Spread environment, we use three homogeneous agents that share a replay buffer and a common policy structure. The learning rate is set to 7e-4, the episode length is set to 25, and the number of landmarks is 3. In the Reference environment, we use two homogeneous agents with the same learning rate, episode length, and number of landmarks.

**SMAC** In the SMAC environment, we set the learning rate to 5e-4, set episode length to 400, and run 2e6 gradient updates in all three environments *1c3s5z*, *MMM*, and *3s_vs_3z*.

For more comprehensive evaluation, we also apply our BPP mechanism to the Independent PPO (De Witt et al., 2020) (IPPO) algorithm, where each agent is treated as an independent PPO learner that optimizes its own policy using only local observations and rewards. As shown in the following Figure 4, the left two figures show the tradeoff between communication interval and the performance, while the right two figures show that our BPP mechanism leads to clear improvements on IPPO with the same communication constraint and further demonstrates the generality of our approach.

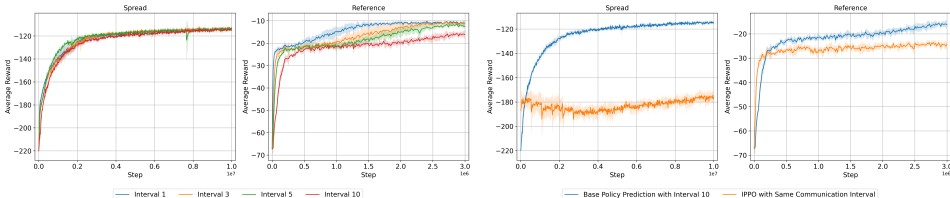

Figure 4: The Left two figures show the comparisons of base policy prediction under different communication intervals. The right two figures show the comparisons between IPPO and base policy prediction.

