# OpenReview forum: "Cooperative Multi-agent RL with Communication Constraints"
_ICLR.cc/2026/Conference — Submitted to ICLR 2026_

### Official Review · Reviewer_arxL · 2025-10-25

**Soundness:** 2
**Presentation:** 1
**Contribution:** 3
**Rating:** 4
**Confidence:** 3

**Summary:**

This paper presents a theoretical framework for cooperative multi-agent reinforcement learning (MARL) under communication constraints by unifying two classes of problems: potential games (PGs) and Markov cooperative games (MCGs). The authors first analyze static potential games where agents’ incentives are aligned through a potential function and propose a communication-efficient algorithm that provably converges to an $\epsilon$-Nash equilibrium with quantified sample and communication complexities. They then extend this framework to the sequential setting of MCGs, where all agents share a common reward, and prove convergence to an agent-wise $\epsilon$-local optimum. The paper provides rigorous definitions, assumptions, and proofs establishing both sample and communication complexity bounds, showing that cooperation can be achieved efficiently despite limited information exchange. The theoretical development is elegant and mathematically sound, offering a clear link between cooperative game structures and decentralized optimization.

**Strengths:**

- The algorithm achieves explicit bounds on both sample and communication complexity, which is an increasingly important issue for large-scale MARL systems.
- The paper rigorously connects Potential Games and Markov Cooperative Games, offering a unifying theoretical lens for cooperative MARL. Provides precise definitions, proof sketches, and clear assumptions

**Weaknesses:**

- The paper claims that base policy prediction can proactively anticipate a sequence of base policies and thereby reduce the number of communication rounds required for new data collection. However, there is no clear analysis or empirical evidence demonstrating that these predictions are accurate enough or that the number of communication rounds is indeed reduced.
- Some implementation details are unclear or inconsistent with the algorithmic description. For instance, the variable $I_t$ in Algorithm 2 is not defined or initialized anywhere in the manuscript, making it difficult to follow.
- Both Algorithm 1 and Algorithm 2 specify adaptive communication triggers, yet the experiments use predefined communication intervals. This inconsistency indicates a gap between the theoretical algorithm and the implemented version.
- In the congestion game experiment, the authors claim that when the communication interval is large, the naive importance sampling baseline becomes outdated and fails to converge. However, since the communication interval is described as fixed, it is unclear how this conclusion is supported.
- The writing and presentation could be improved for clarity. For example, the explanation of Figure 2 (especially regarding MAPPO) is confusing. It is not clear whether MAPPO is a distinct baseline or whether both curves correspond to the proposed method with and without base policy prediction.
- Since the contribution is on communication-efficient, more related work should be discussed regarding communication in cooperative MARL.
- (minor) Figures 2 and 3 are not explicitly referenced in the main text, which disrupts the flow of reading and makes it hard to connect the results to the discussion.

**Questions:**

- Can the authors provide quantitative evidence showing that the proposed base policy prediction reduces the total number of communication rounds compared to standard synchronization schemes?
- The experimental baselines are limited. Since the focus is on communication-efficient learning, have the authors considered comparing with other communication strategies such as the multiple synchronization/communication rules described in [1]? More empirical comparison or comprehensive discussion of communication in MARL in related work will make this work stronger.

#### [1] Hsu et al., "Randomized Exploration in Cooperative Multi-Agent Reinforcement Learning", NeurIPS 2024

---

> ### Author Response · Authors · 2025-11-22
>
> We sincerely appreciate the time and effort you have devoted to reviewing our work. We will address your questions below.
>
> > 1. Empirical evidence that the number of communication rounds is indeed reduced (Weakness 1 and Question 1)
>
> For potential games, note that the standard NPG algorithm does not have any communication constraints, and all agents receive the reward functions for each iteration. Hence, the number of communication rounds is equivalent to the number of iterations $5000$. We add the performance of the standard NPG algorithm with 5000 communication rounds in our Figure 1(a). We can see that our BPP algorithm can achieve similar convergence performance, while only using $5000/500=10$ communication rounds.  For congestion games, we also add similar experiments to show that the BPP algorithm actually keeps comparable performance, while the number of communication rounds is reduced from $500$ to $[500/30]+1=17$. You can see our updated experiment results. We think both these experiments and explanations are empirical evidence of reducing the communication rounds by the BPP algorithm.
>
> For experiments on MAPPO, the empirical evidence for reducing the number of communication rounds appears in the left subfigure of Figure 2 and in Figure 3(a). These results show that our method achieves convergence with a communication interval $I$ from 1 to 10. Recall that the communication interval $I$ indicates that all agents synchronize once every $I$ steps (and each step contains multiple gradient updates). Therefore, when the total number of steps is fixed, using a larger interval $I$ leads to an $I$-fold reduction in the number of communication rounds compared to the classical MAPPO algorithm (which corresponds to $I=1$ in our figures), while achieving comparable performance.
>
> For example, in the MAPPO algorithm on the MPE "spread" environment, all agents perform 10 gradient updates per round when collecting samples for a single step. Thus, MAPPO requires one communication round per step, resulting in 1e7 communication rounds for the whole learning process. In contrast, our method with $I=10$ exhibits similar convergence behavior while synchronizing only once every 10 steps. Consequently, the total number of communication rounds is reduced to 1e6. Hence, the total number of communication rounds can be reduced to 1e6. In other words, the communication interval $I$ means that we reduce the number of communication rounds from 1e7 to $1e7/I$.
>
> We hope this explanation about the empirical results can help you better understand our experiments. We will add more explanations in our revised version.
>
> > 2. Some implementation details are unclear or inconsistent with the algorithmic description.
>
> We set $I_1=1$ for the initialization. The parameter $I_t$ is used for the doubling trick of checking whether we should update the policy, following the V-learning procedure (Algorithm 5 in [10]). We will add more implementation details in Appendix C, and correct some inconsistent notations in our revised version.  Thanks for pointing this out.
>
> > 3. Both Algorithm 1 and Algorithm 2 specify adaptive communication triggers, yet the experiments use predefined communication intervals.
>
> We acknowledge that our practical implementation employs a predefined communication interval, which introduces a gap between the theoretical algorithm and its practical counterpart.
>
> First, we emphasize that the primary contribution of our work lies in the theoretical development. Our analysis breaks the previous upper bounds on communication complexity and demonstrates that, with an adaptive communication trigger, the required number of communication rounds can be significantly reduced. The central idea is the BPP mechanism, which predicts future base policies to improve the efficiency of importance sampling.
>
> Because our paper is theory-oriented, the main purpose of the experiments is to verify that the BPP mechanism is not merely a theoretical construct used to obtain improved bounds, but also provides many benefits in practical settings. For simplicity and ease of implementation, we adopt a fixed communication interval in our experiments to highlight that the BPP mechanism indeed accelerates learning and reduces communication rounds. We believe that using a predefined communication interval is more practical and readily applicable in real-world multi-agent systems.

---

> ### Author Response · Authors · 2025-11-22
>
> > 5. The writing and presentation could be improved for clarity. For example, the explanation of Figure 2 (especially regarding MAPPO) is confusing. It is not clear whether MAPPO is a distinct baseline or whether both curves correspond to the proposed method with and without base policy prediction.
>
> Sorry for the confusion. Since MAPPO is a well-known algorithm in MARL, we did not include additional explanations. In the revised version, we will state clearly that MAPPO is a baseline from previous work, and the orange curve in the rightmost two figures is indeed MAPPO.
>
> > 6. Since the contribution is on communication-efficient, more related work should be discussed regarding communication in cooperative MARL.
>
> We add more discussion of related work here. Communication is a primary mechanism for enabling collaboration among multiple agents. To improve communication efficiency, many prior studies have explored how to reduce the amount of information exchanged in multi-agent RL. Common approaches include parameter sharing and constructing communication networks among agents [2,3,4,5], as well as sharing critic updates across agents [6,7]. Even if they study how to compress the communication, most of these methods still require communication at every iteration.
>
> Paper [7] further reduces communication frequency by broadcasting messages through a learnable message encoder only when the local Q-function exhibits high variance. In contrast, our method does not rely on any message encoder that needs extra training and can therefore be much easier to integrate into existing policy gradient–style algorithms. Moreover, we assume the agent can not even see the local reward functions without communication to the data buffer, making it impossible for agents to estimate local value functions. Last, our BPP mechanism also has a good theoretical guarantee for both performance and the communication cost.
>
> Another line of work [1,8,9] considers reducing communication in a particular multi-agent setting called parallel MDP. We refer the reviewer to our response to Q9 for more discussions.
>
> We will add the discussion above in our revised version to improve our related work section.
>
>
>
> > 7. Figures 2 and 3 are not explicitly referenced in the main text.
>
> Thanks for the reminder. We have added references for our Figures 2 and 3 for more clearer presentation.
>
> > 8. The experimental baselines are limited.
>
> Since our BPP mechanism is built upon the policy gradient style algorithm, we only use the most popular policy gradient style MARL algorithm MAPPO as our baseline. We also add the Independent PPO for additional experiments in the refined Appendix C.
>
> 9. Discussion with [1].
>
> We find that the paper [1] also considers the cooperative MARL, with some analysis for communication complexity. Theoretically, we want to clarify that our settings are much more general than their settings. In their paper, they focus on parallel MDPs, where each agent $i$ corresponds to a different MDP with independent states $s_i$ and action $a_i$, and the reward functions for each agent $i$ only depend on $s_i$ and $a_i$. The only connection of all agents is that they share the same features for a linear MDP. Their communication reducing strategy is highly dependent on this particular structure, since it computes the covariance matrix of the features in the historical data (Eq 3.3 in [1]). Compared to them, our setting is much more general since the rewards for each agent $i$ are dependent on joint action $a = (a_1, \cdots, a_n)$, and all agents also have a global transition kernel $\mathbb{P}(s'\mid s,a)$ dependent on the joint action. This is much more practical for a multi-agent RL benchmark.
>
> Empirically, their communication strategy cannot be applied to our setting and experiments since the strategy highly depends on this particular parallel MDP structure. In fact, with complex environments such as Super Mario in [1], it seems that they do not implement a communication-reducing strategy in their algorithms for this complex environment. Compared to them, we think our MBPP mechanism is much more practical and can be applied to any policy gradient-style algorithm.
>
> Thank you again for your thoughtful and constructive review. We hope this rebuttal can address your concerns and clarify some confusion. We would sincerely appreciate it if you could raise your rating if your concerns are resolved. We are also happy for more discussion.

---

> ### Author Response · Authors · 2025-11-22
>
> [1]. Hsu et al., "Randomized Exploration in Cooperative Multi-Agent Reinforcement Learning", NeurIPS 2024
>
> [2]. Yi et al. "Learning to share in networked multi-agent reinforcement learning." NeurIPS 2022.
>
> [3]. Kim et al. "Communication in multi-agent reinforcement learning: Intention sharing." ICLR 2021.
>
> [4]. Sukhbaatar et al. "Learning multiagent communication with backpropagation." NeurIPS 2016.
>
> [5]. Peng et al. "Multiagent bidirectionally-coordinated nets: Emergence of human-level coordination in learning to play starcraft combat games." preprint 2017.
>
>
> [6]. Chen et al. "Communication-efficient decentralized multi-agent reinforcement learning for cooperative adaptive cruise control." IEEE Transactions on Intelligent Vehicles (2024).
>
> [7].  Zhang et al. "Efficient communication in multi-agent reinforcement learning via variance-based control." NeurIPS 2019.
>
> [8]. Dubey et al. "Provably efficient cooperative multi-agent reinforcement learning with function approximation." preprint (2021).
>
> [9]. Min et al. "Cooperative multi-agent reinforcement learning: Asynchronous communication and linear function approximation." ICML 2023.
>
> [10]. Wang et al. Breaking the curse of multiagency: Provably efficient decentralized multi-agent rl with function approximation. COLT 2023.

---

### Official Review · Reviewer_TaTs · 2025-10-29

**Soundness:** 3
**Presentation:** 3
**Contribution:** 3
**Rating:** 6
**Confidence:** 2

**Summary:**

This paper addresses a critical challenge in cooperative multi-agent reinforcement learning (MARL) — communication constraints that prevent agents from frequently sharing global information (e.g., other agents’ actions or team rewards). Traditional MARL methods typically assume constant communication, which is unrealistic in decentralized systems.

The key idea is to improve learning when communication is infrequent by proposing a method called Base Policy Prediction (BPP).

Instead of using stale base policies in importance sampling (IS) (as in TRPO or PPO, which can cause high variance and instability), the authors propose to predict future base policies using old gradients. This keeps the estimated base policy closer to the current policy, reducing variance and allowing fewer communication rounds without sacrificing performance.

**Strengths:**

Novel Theoretical Advancement:
Introduces Base Policy Prediction, a modification to importance sampling that bridges the gap between outdated and current policies.

Improved Efficiency:
Achieves state-of-the-art results in both communication cost and sample complexity, removing the dependence on the joint action space size.

Strong Theoretical Guarantees:
Provides formal convergence proofs to an ε-Nash equilibrium and clear bounds on communication and sample complexity.

Practical Validation:
Integrates with existing frameworks (such as MAPPO) and demonstrates empirical effectiveness under realistic communication constraints.

**Weaknesses:**

Dependence on Gradient Prediction Accuracy:
The success of Base Policy Prediction heavily depends on the accuracy of old gradient estimates. Noisy or non-stationary environments could degrade performance.

ε-Nash Equilibrium vs. Global Optimum:
The algorithm converges to an ε-Nash equilibrium or local optimum, which is not necessarily the globally optimal cooperative solution.

**Questions:**

I am new to this area, so please forgive any possible misunderstandings.

Does the communication cost for broadcasting the replay buffer bound the total amount of data available to each agent? How is it connected to sample efficiency?

If BPP can reduce the variance of importance sampling and enable reusing old trajectories for more policy updates, could it also improve sampling efficiency for general (non–multi-agent) RL?

---

> ### Author Response · Authors · 2025-11-22
>
> We sincerely appreciate the time and effort you have devoted to reviewing our work. We will address your questions below.
>
>
> > 1. The success of Base Policy Prediction heavily depends on the accuracy of old gradient estimates. Noisy or non-stationary environments could degrade performance.
>
> In this paper, we focus on static environments. Reducing communication in non-stationary environments is much more challenging and may be impossible without additional assumptions on the environments, since information shared during the previous synchronization round can quickly become outdated as the environment evolves. An interesting direction for future work is to study settings in which the environment changes slowly (i.e., where the rate of change is bounded), and to investigate whether communication can be reduced while still maintaining reliable learning performance. We believe our BPP can still have some benefits if the non-stationarity of the environment can be controlled.
>
>
>
>
> > 2. ε-Nash Equilibrium vs. Global Optimum
>
> Our algorithms can only achieve $\varepsilon$-Nash Equilibrium for a potential game. In the general MCG settings, we have shown that $\varepsilon$-Nash Equilibrium is equivalent to an agent-wise local maximum. Achieving the global optimum in an MCG is challenging, as it requires joint optimization over the entire joint action space, which grows exponentially with the number of agents. We argue that the agent-wise local maximum serves as a reasonable proxy for the global optimum because, in many practical multi-agent scenarios, it is a stable point for all agents. There is no way for one agent to deviate from the current policy to improve the team reward.
>
> > 3. Does the communication cost for broadcasting the replay buffer bound the total amount of data available to each agent? How is it connected to sample efficiency?
>
> This is an important question. In this paper, we do not focus on the total amount of data that is exchanged. Instead, we measure communication cost primarily in terms of the number of synchronization rounds, because in practical systems, each synchronization requires multiple agents to establish a communication channel—either with one another or with a centralized buffer. This setup process is typically far slower and more costly than the subsequent exchange of information once the channel is established. Designing methods that further compress the transmitted information and reduce the total communication volume is an interesting and promising direction for future work.
>
> > 4. If BPP can reduce the variance of importance sampling and enable reusing old trajectories for more policy updates, could it also improve sampling efficiency for general (non-multi-agent) RL?
>
> Yes. Our multi-agent RL is a more general setting than the single-agent RL. We choose multi-agent RL because the BPP is much more important here since it can help agents to reduce the communication cost. Our BPP mechanism is also useful in reward-delayed settings or batched settings of single-agent RL, in which the agent can only get access to some data buffer and retrieve the rewards with low frequency.
>
> Thank you again for your positive and constructive review. We hope this rebuttal can address your concerns and clarify some confusion. We would sincerely appreciate it if you could raise your rating if your concerns are resolved. We are also happy to engage in more discussion.

---

### Official Review · Reviewer_xCbV · 2025-10-30

**Soundness:** 4
**Presentation:** 4
**Contribution:** 3
**Rating:** 4
**Confidence:** 4

**Summary:**

This paper tackles the challenge of multi-agent reinforcement learning under limited communication, a realistic but underexplored setting where agents cannot frequently share global information such as rewards or others’ actions due to communication costs.
The authors propose a novel method called Base Policy Prediction, which extends importance sampling by predicting future base policies using old gradients. This technique allows agents to maintain more accurate gradient estimates without frequent synchronization, thereby reducing communication rounds while preserving learning stability.

**Strengths:**

Originality:

The Base Policy Prediction mechanism is a creative modification to classical importance sampling, introducing gradient-based prediction of base policies rather than relying on static ones.
It bridges a clear gap between theoretical MARL under communication constraints and practical distributed implementations like MAPPO, a direction that has been rarely formalized.

Quality:


The sample and communication complexity improvements are significant.
The empirical section validates theory across both toy potential games and complex MARL benchmarks, confirming the benefits in realistic environments.
The comparison table provides a transparent theoretical baseline against prior work.

Clarity:

The presentation of algorithms, changing conditions, and intuition behind communication triggers is commendably clear for a mathematically heavy paper.

Significance:

Addresses a real-world bottleneck in cooperative MARL, which is crucial for scaling to robotics, traffic, and IoT applications.
The framework provides a principled bridge between theory and practice for communication-limited multi-agent learning.

**Weaknesses:**

While the experiments show promising results, the paper could include quantitative comparisons of communication vs. performance trade-offs across a wider range of intervals.

The SMAC and MPE experiments are promising but lack variance/error bars and statistical significance tests to confirm robustness.

The Base Policy Prediction approach requires computing and storing multiple predicted policies per round.


All agents are assumed homogeneous in terms of policy structure and reward access. Real-world MARL often involves heterogeneous agents; it is unclear how well BPP generalizes there.


The connection between the theoretical oracle in Potential Games and the approximate solver used in experiments could be made more explicit.

**Questions:**

How does the algorithm scale with continuous or large discrete action spaces? Could the base policy prediction mechanism be adapted using function approximation to handle such cases efficiently?

Since the BPP relies on predicting gradients from old data, how sensitive is the convergence to gradient noise or non-stationarity in the environment?

Could the base policy prediction mechanism extend to non-cooperative or general-sum settings, possibly with bounded regret guarantees?

The paper mentions the codebase in the supplementary material, but it would be helpful to specify the hyperparameters, communication interval values, and exact network architectures for reproducibility.

---

> ### Author Response · Authors · 2025-11-22
>
> We sincerely appreciate the time and effort you have devoted to reviewing our work. We will address your questions below.
>
>
>
> > 1. The paper could include quantitative comparisons of communication vs. performance trade-offs across a wider range of intervals
>
> In our left subfigure of Figure 2 and Figure 3(a), we show that with a communication interval $I$ from 1 to 10, the performance will get worse. Recall that the communication interval $I$ indicates that all agents synchronize once every $I$ steps (and each step contains multiple gradient updates). Therefore, when the total number of steps is fixed, using a larger interval $I$ leads to an $I$-fold reduction in the number of communication rounds. Hence, it shows a tradeoff between the number of communications and the performance.
>
> > 2. The SMAC and MPE experiments are promising but lack variance/error bars and statistical significance tests to confirm robustness.
>
> Figures 2 and 3 include variance bars computed over 5 random seeds, where the shaded regions represent the standard deviation. We believe it is sufficient to demonstrate the robustness of our results.
>
> > 3. The Base Policy Prediction approach requires computing and storing multiple predicted policies per round.
>
> Yes, we acknowledge that our approach requires computing and storing multiple predicted policies. However, we view this as a necessary tradeoff between communication efficiency and memory/computation efficiency. By storing several predicted policies locally, we can substantially reduce the required communication rounds, which is highly valuable in settings where establishing a communication channel to the data buffer is expensive. For example, in a large robotic system for package transportation, communication typically depends on network access to a centralized data buffer, which can be slow or unreliable. In contrast, equipping each robot with enough memory to store a few predicted policies is relatively easy.
>
> > 4. All agents are assumed homogeneous in terms of policy structure and reward access.
>
> All agents have the same policy structure and access the reward in the same way is a common assumption in many empirical multi-agent RL benchmarks such as MPE and SMAC, as well as in theoretical MARL frameworks [1,2]. Addressing heterogeneity among agents is an exciting and important open problem. We also note that even when agents are homogeneous, they can still receive different rewards and exhibit distinct behavioral patterns, since the reward depends on the joint action of all agents, and each agent should behave differently to achieve the best performance.
>
> > 5. How does the algorithm scale with continuous or large discrete action spaces? Could the base policy prediction mechanism be adapted using function approximation to handle such cases efficiently?
>
> For the continuous action spaces, we can use linear function approximation as a solution. Some previous linear function approximation on MARL considers $r(s,a)=\theta^\top \phi(s,a)$ for the joint action $a$. However, assuming one agent knows the $\phi(s,a)$ for any joint action $a$ is too strong and leads the MARL setting more like a single-agent MDP. Hence, we formulate $r_i(s,a)=\theta(s,a_{-i})^\top  \phi(s,a_{i})$, and the marginalized reward $\ell_i (a_i, \pi) = \mathbb{E}\_{a_{-i}\sim \pi} [r_i(s,a)] = \mathbb{E}\_{a_{-i}\sim \pi}[\theta(s,a_{-i})]^\top \phi(s,a_{i})$, which is similar to the decentralized linear function approximation in [1]. We believe that our algorithms can also extend to the linear setting. In particular, we can parameterize both the marginalized reward and the policy class using a linear softmax parameterization, and the counting argument still holds with some assumptions similar to Assumption 4.1. For each communication round, when the current policy $\pi$ is fixed,  instead of sampling any possible action $a_i$ for agent $i$, we can just estimate $\mathbb{E}\_{a_{-i}\sim \pi}[\theta(s,a_{-i})]$ by linear regression. This approach may allow us to remove the explicit dependence on the action space in our final bounds, both in potential games and even in the MCG setting.
>
> > 6. Since the BPP relies on predicting gradients from old data, how sensitive is the convergence to gradient noise or non-stationarity in the environment?
>
> In this paper, we focus on static environments. Reducing communication in non-stationary environments is much more challenging and may be impossible without any assumptions on the environments, since information shared during the previous synchronization round can quickly become outdated as the environment evolves. An interesting direction for future work is to study settings in which the environment changes slowly (i.e., where the rate of change is bounded), and to investigate whether communication can be reduced while still maintaining reliable learning performance. We believe our BPP can still have some benefits if the non-stationarity of the environment can be controlled.

---

> ### Author Response · Authors · 2025-11-22
>
> > 7. non-cooperative or general-sum settings
>
> First, we note that in general-sum games, computing a Nash equilibrium is computationally hard, and exponential sample complexity is unavoidable [3]. Some previous work has shown that if each agent treats the others as adversaries and uses adversarial bandit algorithms to optimize its own reward, this approach can converge to a coarse correlated equilibrium (CCE) [1].
>
> In contrast, our BPP mechanism builds upon the policy gradient–style algorithms with importance sampling, which is very different from such adversarial bandit approaches in non-cooperative or general-sum settings. In order to apply BPP in general-sum settings, we think the first step is to establish a policy gradient-style algorithm for a general-sum game.
>
> > 8. Code
>
> Thanks for the reminder. We will provide more training details in our final version.
>
> Thank you again for your thoughtful and constructive review. We hope this rebuttal can address your concerns and clarify some confusion. We would sincerely appreciate it if you could raise your rating if your concerns are resolved. We are also happy to engage in more discussion.
>
>
>
>
>
> [1]. Wang et al. Breaking the curse of multiagency: Provably efficient decentralized multi-agent rl with function approximation. COLT 2023.
>
> [2]. Xiong et al. "Sample-efficient multi-agent rl: An optimization perspective." ICLR 2024.
>
> [3]. Daskalakis et al. "On the complexity of approximating a Nash equilibrium." TALG 2013.

---

### Official Review · Reviewer_pLby · 2025-11-03

**Soundness:** 3
**Presentation:** 3
**Contribution:** 3
**Rating:** 6
**Confidence:** 4

**Summary:**

The paper studies cooperative multi-agent RL under limited communication and proposes base policy prediction, a modified importance-sampling scheme that precomputes a sequence of base policies from old gradients. This keeps importance weights stable, enabling far fewer communication rounds. The authors prove convergence to $\varepsilon$-Nash equilibria in potential games and extend the framework to Markov cooperative games with polynomial sample complexity.

**Strengths:**

1. The core idea-predicting future base policies via natural-policy-gradient-style updates and collecting data for that whole predicted set in a single communication round-is a clean variance control mechanism for importance sampling under staleness, and it is explicitly tied to the communication trigger condition.

2. The analysis for potential games gives simultaneous bounds on communication rounds $O\left(\varepsilon^{-3 / 4}\right)$ and samples $O\left(\operatorname{poly}\left(\max _i\left|A_i\right|\right) \varepsilon^{-11 / 4}\right)$, improving prior $\varepsilon$-dependence while avoiding exponential dependence on the joint action space.

3. The extension to Markov cooperative games shows the PG oracle can be plugged into a V-learning style outer loop, preserving the communication-saving idea while still yielding an $O(\varepsilon)$ agent-wise local maximum, which demonstrates the method is not limited to the static PG setting.

**Weaknesses:**

1. The communication-trigger condition (two-part test on reward-difference and on elapsed steps) is chosen to bound IS variance, but the paper does not give a tight or adaptive rule showing this condition is near-optimal for a given environment.

2. The sample complexity still carries a relatively high exponent $\varepsilon^{-11 / 4}$; although better than some baselines, the proof does not clarify whether this exponent is an artifact of handling multiple predicted policies or is information-theoretically necessary.

3. The MCG extension relies on Assumption 4.1 (gap and coverage) for every constructed PG instance; this is a strong structural requirement, and the paper does not discuss how sensitive the guarantees are if the gap is small or state-dependent.

**Questions:**

1. In the base-policy prediction step, can the authors quantify the maximal KL (or total-variation) drift between a predicted base policy and the actual future policy that still keeps the IS weights uniformly bounded?

2. The changing condition mixes a value-difference test and a hard cap $t^{\prime} \geq c \varepsilon^{-1 / 4}$. Is there a way to remove the hard cap and trigger purely on variance/proxy estimates while preserving the $O\left(\varepsilon^{-3 / 4}\right)$ communication bound?

3. For the MCG algorithm, the PG oracle is called at multiple stages along the horizon. How does error from approximate PG solutions accumulate across stages, and can it be localized so the overall gap stays $O(\varepsilon)$ without tightening each oracle call?

4. The analysis avoids exponential $\prod_i\left|A_i\right|$ factors by counting distinct predicted policies. Can the same counting argument be extended to continuous action spaces via smoothing or entropy regularization, or is discreteness essential to keep $\left|\Pi_t\right|$ small?

---

> ### Author Response · Authors · 2025-11-22
>
> We sincerely appreciate the time and effort you have devoted to reviewing our work. We will address your questions below.
>
>  > 1. The paper does not give a tight or adaptive rule showing this condition is near-optimal for a given environment, and the proof does not clarify whether this exponent is an artifact of handling multiple predicted policies or is information-theoretically necessary.
>
> This is an important question. However, we want to claim that even for potential game without any communication constraints, the lower bound is only $\Omega(\sum_{i=1}^n |A_i|/\varepsilon^2)$, which is not tight compared to previous upper bound $O(\sum_{i=1}^n |A_i|/\varepsilon^3)$ or $O(\prod_{i=1}^n |A_i|/\varepsilon^2)$. Though our algorithm bridges this gap by providing a tighter $O(\sum_{i=1}^n |A_i|/\varepsilon^{11/4})$ upper bound, it is still an open problem to achieve a tight upper bound for the general potential game, even if without communication constraints. Hence, we believe that providing tight lower bound guarantees with communication constraints is also a hard problem.
>
> > 2. The MCG extension relies on Assumption 4.1 (gap and coverage) for every constructed PG instance; this is a strong structural requirement, and the paper does not discuss how sensitive the guarantees are if the gap is small or state-dependent.
>
> For the sensitivity, our guarantees show that our sample complexity and the error are proportional to $O(1/\Delta)$ and $O(1/(c\Delta)^2)$, respectively. Note that this kind of assumption is unavoidable for NPG-style algorithms in MARL, since some ill conditions will lead to exponential iterations to converge to NE even in single-agent settings for policy
> gradient methods [2].
>
> For General MCG, note that now the $\ell_{i}(s,a)$ for $h$-step is exactly the Q-function $r(s,a) + \mathbb{P}\_{h+1}\overline{V}_{h+1}(s)=\overline{Q}_h(s,a)$. Hence, you can regard the assumption in MCG to be a general version of the Q-functions instead of the reward function in a potential game. We do not think it is a very strong structural requirement. In fact, in MARL literature, it is common to have some structural assumptions on Q-functions. (VDN approach [3], mean-field RL [4], etc, linear function approximation [5].) We also believe making assumptions for multiple stages $h \in [H]$ is necessary for a finite-horizon MDP, since we need to control each stage to get the final error.
>
>
>
> Moreover, we can use the same technique as in [1], to change our assumption to the limits of $c$ and $\Delta$ being lower bounded by some constant. Then, this assumption can be regarded as a local guarantee only for the policies around the final optimal policy, which is not a strong structural requirement.
>
> > 3. In the base-policy prediction step, can the authors quantify the maximal KL (or total-variation) drift between a predicted base policy and the actual future policy that still keeps the IS weights uniformly bounded?
>
> For the predicted base policy $\overline{\pi}^t$ and the actual future policy $\hat{\pi}^t$,  we know that $\overline{\pi}\_i^t = \pi_0^t \cdot \exp(\eta t\hat{\ell}\_i(a,\pi^0))$ with some redistribution on the probability mass, and $\hat{\pi}\_i^t = \pi_0^t \cdot \exp(\eta \sum_{j=0}^{t-1}\hat{\ell}\_i(a,\hat{\pi}^j))$. Based on our changing condition, we know that $|\eta t \hat{\ell}\_i(a,\pi^0)-\eta \sum_{j=0}^{t-1}\hat{\ell}\_i(a,\hat{\pi}^j)| = \widetilde{O}(1/n)$. Hence, by our Lemma A.1, the TV distance $TV(\overline{\pi}_i^t \|\hat{\pi}^t_i) = \widetilde{O}(1/n)$, and then $TV(\overline{\pi}^t\| \hat{\pi}^t)$  is bounded by some constant.
>
> > 4. Is there a way to remove the hard cap and trigger purely on variance/proxy estimates?
>
> The changing condition depends on both the variance estimates and the hard time constraint, which are essential for our theoretical guarantees. In particular, under the first variance–estimation condition, the estimation error also affects $\hat{\ell}$,  and this error accumulates across timesteps. Therefore, restricting the size of the time window helps control the overall estimation error in the variance estimate used by the first changing condition.

---

> ### Author Response · Authors · 2025-11-22
>
> > 5. For the MCG algorithm, the PG oracle is called at multiple stages along the horizon. How does the error from approximate PG solutions accumulate across stages?
>
> The error will accumulate across stages in a linear way. That means if we can get $O(\varepsilon)$ error for one stage, the total error is also bounded by $O(H\varepsilon)$. Therefore, the result of the MCG algorithm has some terms polynomial in $H$. For theoretical proof, you can check Line 1355 for more details.
>
> > 6. Can the same counting argument be extended to continuous action spaces?
>
> First, we want to clarify that avoiding the dependence on $\prod_{i=1}^n |A_i|$ is because of the benign landscape of NPG-style algorithms on potential games, not the counting argument (Line 267). The benefit of this counting argument is to reduce the sample complexity to $O(\varepsilon^{-11/4})$. If we do not use counting argument and just naively bound $|\Pi_t|$ as $T$, then for each communication round it requires $|\Pi_t| \cdot O(\varepsilon^{-2}) = O(T\varepsilon^{-2})=O(\varepsilon^{-3})$ samples (given $T=O(\varepsilon^{-1})$), and the total sample complexity can only bounded by $O(\varepsilon^{-3})\cdot \\#\\{\text{communication rounds}\\}=O(\varepsilon^{-15/4}).$
>
> For the continuous action spaces, we can use linear function approximation as a solution. Some previous linear function approximation on MARL considers $r(s,a)=\theta^\top \phi(s,a)$ for the joint action $a$. However, assuming one agent knows the $\phi(s,a)$ for any joint action $a$ is too strong and reduces the MARL setting to essentially be like a single-agent MDP. Hence, we formulate $r_i(s,a)=\theta(s,a_{-i})^\top  \phi(s,a_{i})$, and the marginalized reward $\ell_i (a_i, \pi) = \mathbb{E}\_{a_{-i}\sim \pi} [r_i(s,a)] = \mathbb{E}\_{a_{-i}\sim \pi}[\theta(s,a_{-i})]^\top \phi(s,a_{i})$, which is similar to the decentralized linear function approximation in [5]. We believe that our algorithms can also extend to the linear setting. In particular, we can parameterize both the marginalized reward and the policy class using a linear softmax parameterization, and the counting argument still holds with some assumptions similar to Assumption 4.1. For each communication round, when the current policy $\pi$ is fixed,  instead of sampling any possible action $a_i$ for agent $i$, we can just estimate $\mathbb{E}\_{a_{-i}\sim \pi}[\theta(s,a_{-i})]$ by linear regression. This approach may allow us to remove the explicit dependence on the action space in our final bounds, both in potential games and even in the MCG setting.
>
> Thank you again for your thoughtful and constructive review. We hope this rebuttal can address your concerns and clarify some confusion. We would sincerely appreciate it if you could consider raising your rating if your concerns are resolved. We are also happy to engage in more discussion.
>
>
>
> [1]. Sun et al. Provably fast convergence of independent natural policy gradient for markov potential games. NeurIPS 2023.
>
> [2]. Li et al. Softmax policy gradient methods can take exponential time to converge. COLT 2021.
>
> [3]. Sunehag et al. Value-decomposition networks for cooperative multi-agent learning. preprint 2017.
>
> [4]. Yang et al. Mean field multi-agent reinforcement learning. ICML 2018.
>
> [5]. Wang et al. Breaking the curse of multiagency: Provably efficient decentralized multi-agent rl with function approximation. COLT 2023.

---

> > ### Comment · Reviewer_pLby · 2025-11-27
> >
> > Thank you for the detailed response. After reading the response, most of my concerns are solved, but I still have some questions and suggestions.
> >
> > W1: I see. While acknowledging the difficulty of a tight lower bound, could you provide intuition on whether the $O(\epsilon^{-11/4})$ complexity is primarily dominated by the variance control mechanism or the base policy prediction steps?
> >
> > W2: (Merged with W1)
> >
> > W3: Thank you for the detailed response, I understand. A further question: you mentioned the assumption can be relaxed to a local guarantee around the optimal policy. Does this imply the algorithm requires a warm-up or initialization to ensure it falls within this local region?
> >
> > Q1: My question has been solved.
> >
> > Q2: I understand the necessity of the hard cap in theory. A small further question: in the experiments, how often is the communication triggered by the variance condition versus the hard time constraint?
> >
> > Q3: I understand, thank you.
> >
> > Q4: The author's discussion and analysis on using function approximation makes sense to me. I think it is an interesting direction.
> >
> > Additional comment: In related work, cooperative MARL is mainly discussed in field of Markov Games. I suggest adding a broader discussion on cooperative MARL literature including centralized setting.
> >
> > If my questions can be fully addressed, I will feel more postive about this paper.

---

> > > ### Author Response · Authors · 2025-12-03
> > >
> > > Thanks for the response and the following questions! We will address your questions below.
> > >
> > >
> > > > 1. Could you provide intuition on whether the $O(\epsilon^{-11/4})$ complexity is primarily dominated by the variance control mechanism or the base policy prediction steps?
> > >
> > > The $O(\epsilon^{-11/4})$ complexity arises from both the variance control mechanism and the base policy prediction steps. To see this, suppose the last communication round is indexed by $0$. If we rely solely on base-policy prediction without enforcing condition (A), one can show that at round $t$: $\phi(\hat{\pi}^{t-1}) - \phi(\hat{\pi}^0)
> > > \ge \widetilde{O}(1/t^3) - O(t\varepsilon),$ as stated in Line 917. The $t\varepsilon$ term arises from the cumulative effect of base policy prediction error and the gradient estimate error over iterations. The hard constraint (condition (B)) then can lead to $\phi(\hat{\pi}^{t-1}) - \phi(\hat{\pi}^0)
> > > \ge \widetilde{O}(1/t^3)$, and finally lead to the final $O(\epsilon^{-11/4})$ complexity. Hence, the rate is not dominated solely by the variance-control mechanism or the base-policy prediction step.
> > >
> > >
> > > > 2. You mentioned the assumption can be relaxed to a local guarantee around the optimal policy. Does this imply the algorithm requires a warm-up or initialization to ensure it falls within this local region?
> > >
> > > Yes. Since the assumption can be relaxed to a local guarantee, a warm-up or initialization is good for the convergence. In our SMAC experiment, we find that an initialization with more frequent communication can avoid the cold-start issue and finally lead to a good performance.
> > >
> > > > 3. In the experiments, how often is the communication triggered by the variance condition versus the hard time constraint?
> > >
> > > In our experiments, we do not apply our theoretical variance condition since it could be difficult to implement in complex environments. Instead, for simplicity and ease of implementation, we adopt a fixed communication interval in our experiments to highlight that the BPP mechanism indeed accelerates learning and reduces communication rounds. Hence, in our experiments, communication was triggered only by the hard time constraint.
> > >
> > > The main purpose of the experiments is to verify that the BPP mechanism also provides benefits and reduces communication in practical settings. Hence, we believe that using a predefined communication interval with BPP mechanism is more practical and readily applicable in real-world multi-agent systems.
> > >
> > > > 4.  In related work, cooperative MARL is mainly discussed in field of Markov Games. I suggest adding a broader discussion on cooperative MARL literature including centralized setting.
> > >
> > > Thanks for the reminder. We have added some discussions about cooperative MARL with communication in our Appendix D.

---

### Meta-Review · Area_Chair_JpcV · 2026-01-06

**Summary:**

This paper studied cooperative multi-agent reinforcement learning with limited communication and communication costs. The paper proposed a new method based on "predicting" a sequence of base policies based on old gradients, which reduces the mismatch between the importance-sampling base and the current policy, and thus enables sample collection for multiple predicted bases within a single communication round. The communication complexity savings were then proved for potential games and the extension of Markov cooperative games. The paper contains new algorithmic ideas, with overall solid theoretical contributions. I especially like the implication and connection to practical algorithms like MAPPO. However, as a theoretical paper, there were some concerns regarding the tightness of the bounds, the restrictiveness of certain assumptions, and the gap between theory and experiments. From my reading, although the "algorithm/method" is novel, the technical novelty of the "analysis" appeared relatively limited as well. Overall, it reached a consensus that this is a borderline paper.

**Reviewer Concerns:**

The concerns regarding the clarifications of the theoretical intuition, the experimental interpretation, and presentation/positioning issues were mostly addressed. However, the gap between theory and experiments, the tightness of the theoretical results, and the breadth of the baseline algorithms are still relatively standing.

**Reviewer Scores:**

Reviewer pLby would have probably increased the score by 1, as they explicitly commented that "most of my concerns are solved" and feel “more positive” if the remaining questions were addressed. It was not very clear if other reviewers would have increased their scores.

---

### Decision · Program_Chairs · 2026-01-26

Reject